# Interactive rank testing by betting

**Boyan Duan**                                                        BOYAND@STAT.CMU.EDU
**Aaditya Ramdas**                                                  ARAMDAS@STAT.CMU.EDU
**Larry Wasserman**                                                    LARRY@STAT.CMU.EDU
*Department of Statistics and Data Science, Carnegie Mellon University, Pittsburgh, PA 15213*

**Editors:** Bernhard Schölkopf, Caroline Uhler and Kun Zhang

## Abstract

In order to test if a treatment is perceptibly different from a placebo in a randomized experiment with covariates, classical nonparametric tests based on ranks of observations/residuals have been employed (eg: by Rosenbaum), with finite-sample valid inference enabled via permutations. This paper proposes a different principle on which to base inference: if — with access to all covariates and outcomes, but without access to any treatment assignments — one can form a ranking of the subjects that is sufficiently nonrandom (eg: mostly treated followed by mostly control), then we can confidently conclude that there must be a treatment effect. Based on a more nuanced, quantifiable, version of this principle, we design an *interactive* test called i-bet: the analyst forms a single permutation of the subjects one element at a time, and at each step the analyst *bets* toy money on whether that subject was actually treated or not, and learns the truth immediately after. The wealth process forms a real-valued measure of evidence against the global causal null, and we may reject the null at level $\alpha$ if the wealth ever crosses $1/\alpha$. Apart from providing a fresh "game-theoretic" principle on which to base the causal conclusion, the i-bet has other statistical and computational benefits, for example (A) allowing a human to adaptively design the test statistic based on increasing amounts of data being revealed (along with any working causal models and prior knowledge), and (B) not requiring permutation resampling, instead noting that under the null, the wealth forms a nonnegative martingale, and the type-1 error control of the aforementioned decision rule follows from a tight inequality by Ville. Further, if the null is not rejected, new subjects can later be added and the test can be simply continued, without any corrections (unlike with permutation p-values). Numerical experiments demonstrate good power under various heterogeneous treatment effects. We first describe i-bet test for two-sample comparisons with unpaired data, and then adapt it to paired data, multi-sample comparison, and sequential settings; these may be viewed as interactive martingale variants of the Wilcoxon, Kruskal-Wallis, and Friedman tests.

**Keywords:** Covariate-Adjusted Wilcoxon; Interactive Rank Tests; Randomized Experiments.

## 1. Introduction

The problem of testing whether a treatment has any effect in a randomized experiment without parametric assumptions is frequently encountered in biology, medical research, and social sciences (see, for example, Olive et al. (2009); Aguilera et al. (2017); Rastinehad et al. (2019)). A classical nonparametric method is the Wilcoxon test, and there have been several proposed extensions that adjust for covariates, in order to better detect the treatment effect. For example, suppose we want to evaluate a medication by comparing the blood pressure (outcome) of subjects who take the medication (treatment) with that of subjects who do not (control). The blood pressure could be affected by the subject's gender, age, etc.—accounting for these would help increase power, especially when the medication only affects a subpopulation. In this paper, we use a novel "game-theoretic" principle of

guessing and betting on the treatment assignments using all data except the truth assignments, and conclude there is an effect if most guesses are correct. Our proposed test is "interactive" — it allows an analyst to look at (progressively revealed) data and adaptively explore arbitrary working models for making the bets — which improves power especially under heterogeneous treatment effects.

## 1.1. Problem setup

Consider a sample with $n$ subjects. Let the outcome of subject $i$ be $Y_i$, the covariates be $X_i$, and the treatment assignments be indicators $A_i$ for $i \in [n] \equiv \{1, 2, \ldots, n\}$. The null hypothesis of interest is that there is no difference between treated and control outcomes conditional on the covariates [1]:

$$H_0 : (Y_i \mid A_i = 1, X_i) \stackrel{d}{=} (Y_i \mid A_i = 0, X_i) \text{ for all } i \in [n]. \tag{1}$$

Rejecting the above null means that there exist some subjects who respond differently when treated or not. We do not further identify which subject respond differently. Testing the above global null may appear in an exploratory analysis to see whether the treatment has any effect on any person, or as a building block within a closed testing procedure. For our interactive algorithm that we propose later to succeed in rejecting the global null, it must indeed *implicitly* learn which part of the covariate space exhibits this difference between treatment and placebo, and if the global null is rejected, one may use this information to design follow-up studies or analyses focused on other goals.

This paper deals with classic randomized experiments, and in particular we assume that

(i) (random assignment) the treatment assignments are independent and randomized:

$$\mathbb{P}(A_i = 1 \mid X_i) = \mu_i \equiv \mu_i(X_i) \in (0, 1), \text{ where } \mu_i \text{ is known for all } i \in [n];$$

(ii) (no interference) the outcome of any subject $Y_{i_1}$ depends only on their assignment $A_{i_1}$ and does not depend on the assignment $A_{i_2}$ for any $i_1 \neq i_2 \in [n]$.

Note that the above considers the case where the probabilities of receiving treatment $\{\mu_i(X_i)\}_{i=1}^n$ are known (but the total number of treated subjects is not fixed). The methods are easily extended to the case where the number of treatments is fixed: $\sum_{i=1}^n A_i = m$, and they are assigned to a random subset of subjects (see Remark 5).

To enable us to effectively adjust for covariates, we use the following "working model":

$$Y_i = \Delta(X_i)A_i + f(X_i) + U_i, \tag{2}$$

where $\Delta(X_i)$ is the treatment effect, $f(X_i)$ as the control outcome, and $U_i$ is zero mean 'noise' (unexplained variance). When working with such a model, we effectively want to detect if $\Delta(X_i)$ is nonzero. Importantly, model (2) only exists on the analyst's computer, and it need not be correctly specified or accurately reflect reality in order for the tests in this paper to be valid (but the more ill-specified or inaccurate the model is, the more test power may be hurt).

---

1. An alternative, equivalent description is that each subject $i$ has potential control outcome $Y_i^C$, potential treated outcome $Y_i^T$, and the treatment indicator $A_i$ for $i \in [n] \equiv \{1, 2, \ldots, n\}$. The observed outcome is $Y_i = Y_i^C(1 - A_i) + Y_i^T A_i$ under the standard causal assumption of consistency ($Y_i = Y_i^T$ when $A_i = 1$ and $Y_i = Y_i^C$ when $A_i = 0$). In this setting, the potential treated outcome $Y_i^T$ corresponds to $Y_i | A_i = 1$.

### 1.2. Rosenbaum's covariance-adjusted Wilcoxon test

Recall that the Wilcoxon rank-sum test (also referred to as the Mann–Whitney U-test) calculates

$$W^{\text{ori}} = \sum_{i=1}^{n} (2A_i - 1) \operatorname{rank}(Y_i),$$

where $\operatorname{rank}(Z_i)$ is the rank of $Z_i$ amongst $\{Z_i\}_{i=1}^{n}$. When the treatment effect is large, the subjects receiving treatment ($A_i = 1$) tend to have larger outcomes, and hence $W^{\text{ori}}$ would be large. Rank-based statistics have been explored in many directions: see Lehmann and D'Abrera (1975) for a review. Recent work focuses on how to incorporate covariate information to improve power. Zhang et al. (2012) develop an optimal statistic to detect constant treatment effect; in multi-sample comparison, Ding and Keele (2018) numerically compare rank statistics of outcomes or residuals from linear models; Rosenblum and Van Der Laan (2009) and Vermeulen et al. (2015) focus on related testing problems for conditional average effect and marginal effect; Rosenbaum (2010) and Howard and Pimentel (2020) use generalizations of rank tests for sensitivity analysis in observational studies. Here, we focus on improving power under heterogeneous treatment effects.

Rosenbaum (2002) proposed the covariance-adjusted Wilcoxon test that considers the residuals of regressing the outcome $Y_i$ on covariates $X_i$ (without assignment $A_i$). Specifically, denote the residual for subject $i$ as $R_i$:

$$R_i \equiv R_i(Y_i, X_i) := Y_i - \widehat{Y}(X_i), \tag{3}$$

where $\widehat{Y}(X_i)$ the prediction of $Y_i$ using $X_i$ via any modeling and $R_i$ can be viewed as an approximation of the treatment effect after accounting for heterogeneous control outcome. The covariance-adjusted Wilcoxon test replaces the outcomes with the residuals:

$$W^{\text{CovAdj}} = \sum_{i=1}^{n} (2A_i - 1) \operatorname{rank}(R_i), \tag{4}$$

abbreviated as CovAdj Wilcoxon test in the rest of the paper. The rejection rule is based on permutation. Note that under the null, the assignment $A_i$ is independent of other data information $\{Y_i, X_i\}$. The permutation test estimates the null distribution of $W$ by permuting the treatment assignments $\{A_i\}_{i=1}^{n}$, described as follows:

**Input:** Outcomes, treatment assignments, and covariates $\{Y_i, A_i, X_i\}_{i=1}^{n}$, target Type-I error rate $\alpha$;
**Procedure:** 1. Calculate $W$ using the observed data $\{Y_i, A_i, X_i\}_{i=1}^{n}$;
2. Let $W^1 = W$ and for $b = 2, \ldots, B$, generate a random permutation of the treatment assignments $(A_1^b, \ldots, A_n^b)$; and calculate $W^b$ using the permuted data $\{Y_i, A_i^b, X_i\}_{i=1}^{n}$;
3. Let $W^{(j)}$ be the $j$-th largest order statistic. Reject the null if $W > W^{(\lfloor \alpha \cdot B \rfloor)}$.
 **Algorithm 1:** Framework for the permutation test

The signed-rank test offers a general formula to construct permutation tests with various forms of test statistics $W$ for two-sample comparison, which we discuss in Appendix G.

### 1.3. Interactively constructing a ranking, and betting on it

In contrast to one-step tests such as CovAdj Wilcoxon test, we propose a multi-step test that progressively guesses and bet on the treatment assignments. The intuitive idea is that if based on all covariates and outcomes, a human analyst can guess most treatment assignments correctly, then the treatment must have an effect and we can reject the null. Several advantages of taking the above betting perspective include: (a) the flexibility for the analyst to use combine (partial) data, prior knowledge, and arbitrary modeling for guessing and betting on the treatment assignments; (b) the bets are used to construct a sequence of test statistics and form a multi-step protocol, during which the analyst can monitor the current algorithm's performance and is allowed to make adjustments to their working model at any step; (c) the constructed test statistics form a nonnegative martingale, and the type-I error control follows from a martingale property (detailed later), avoiding the high computation cost in data permutation for the rejection rule. Despite allowing human interaction in (a) and (b), the proposed test always maintains valid type I error control without assuming any working model specified by the analyst to be correct.

Our proposed test by betting can be viewed as separating the information used for betting and interactive algorithm design and that for testing, via "masking and unmasking" (Figure 1). Masking means we hide the information of treatment assignments $\{A_i\}_{i=1}^n$ from the analyst. Unmasking refers to the process of revealing the masked assignments one at a time to the analyst. Consider a simple case where the treatment is assigned to each subject independently with $1/2$ probability. The test considers the cumulative products

$$M_t = \prod_{j=1}^t \left[1 + w_j \cdot (A_{\pi_j} - 1/2)\right],\tag{5}$$

where $\{\pi_j\}_{j=1}^n$ denotes an ordering interactively decided by the analyst, and $w_j \in [-2, 2]$ is a user-defined bet on the treatment assignment, both of which can be based on all the revealed information $\{Y_i, X_i\}_{i=1}^n$ and the true treatment assignments of all previous subjects in the ordering $A_{\pi_1}, \ldots, A_{\pi_{j-1}}$. Regardless of the specific choices of the ordering and binary estimations and weights, $w_j \cdot (2\widehat{A}_{\pi_j} - 1)$ is independent of $A_{\pi_j}$ under the null (but not under the alternative); and thus, the process $(M_t)$ is a nonnegative martingale under the null. We reject the null as soon as $M_t \geq 1/\alpha$ for some $t \in [n]$ (see the precise description of our procedure in Algorithm 2 of Section 2). Type-1 error control is guaranteed by Ville's (often attributed to Doob) maximal inequality (Ville, 1939), which states that with probability $1 - \alpha$, a nonnegative martingale with initial value one (which $M_t$ is, under the null) will never exceed $1/\alpha$:

$$\Pr_{H_0}(\exists t \geq 1 : M_t \geq 1/\alpha) \leq \alpha.\tag{6}$$

For a self-contained proof of this fact, see (Howard et al., 2020). Ville's inequality holds with equality for continuous path martingales (only possible in continuous time) and in discrete time, the only looseness is due to "overshoot" and is typically negligible (Howard et al., 2020), meaning that the inequality *almost* holds with equality, which is important so that the test is not too conservative.

**Remark 1** *There is nothing special about the use of $1/2$ above, and one can simply use $\mu$ if there is some other probability of random assignment. Further, if subject $i$ was randomized with probability $\mu_i \equiv \mu(X_i)$ possibly depending on its covariates, or in some kind of stratified manner, then we can*

*simply use that $\mu_i$ in (5) instead of $1/2$, retaining the required martingale property; we return to this formally later. (In this case, the range of the bet $w_j$ must be adjusted to $[-1/(1 - \mu_i), 1/\mu_i]$ to ensure nonnegativity.) It appears that in the last case, the exchangeability amongst subjects has been destroyed and so the permutation test in Algorithm 1 does not directly apply — this can be addressed using a recent sophisticated concept called weighted exchangeability that has utility in other settings (Tibshirani et al., 2019), but this leads us far astray; our point is simply that our procedure retains its simplicity under more complex randomization schemes.*

**Remark 2** *Though we do not necessarily recommend viewing i-bet in this way, it is possible to view our interactive test as a computational shortcut for the permutation framework. In short, whenever the test statistic is a nonnegative martingale (by design), permutations can be avoided. To elaborate, one could imagine calibrating the test statistic $W = \max_{1,...,n} M_t$ using algorithm 1. What Ville's inequality implies is that the $\alpha$-upper quantile of the permutation null distribution for $W$ (obtained by permuting the data and construct $\{W^b\}_{b=1}^B$, with $B = n!$ for example), will be at most $1/\alpha$.*

The above test retains validity amidst significant flexibility. For example, the analyst could employ any probabilistic working model or predictive machine-learning algorithm to guess the treatment assignments $\widehat{A}_i \in \{0, 1\}$, perhaps along with an associated level of confidence such as a posterior probability or a score $\nu_i \in [0, 1]$, for each subject $i$ that have not yet been included in the ordering. Then, at step $t$ of the algorithm, the next subject in the ordering $\pi_t$ could be the one where the analyst is most confident, and $w_t$ could equal $2(2\widehat{A}_t - 1) \in \{-2, 2\}$ or $2(2\nu_t - 1) \in [-2, 2]$.

Intuitively, our algorithm tests whether there exist any non-nulls by examining whether we can succeed at guessing the treatment assignments better than random chance, and this is reflected by our ability to form "smart, nonrandom" bets that cause $M_t$ to grow faster than a martingale, and cross $1/\alpha$ as soon as possible. When all subjects are nulls and we have that the treatment assignments are independent of the outcomes and covariates, we cannot distinguish subjects who are treated or not based on the outcomes and covariates, and no algorithm can result in $M_t$ losing this martingale property (and thus having controlled growth). In contrast, if the null is false, we hope that our algorithm will be able to correctly guess the assignments, especially for subjects we are most confident about (ordered upfront), so that the cumulative products $(M_t)_{t=1}^n$ grow large.

Interaction enters in the process of unmasking. Intuitively, to construct large $M_t$ and reject the null, the analyst should guess whether a subject receives treatment while the assignments $\{A_i\}_{i=1}^n$ are hidden. She can guess the treatment assignments using the revealed data information $\{Y_i, X_i\}_{i=1}^n$ and $\{A_{\pi_j}\}_{j=1}^{t-1}$ (for the $t$-th iteration), and any prior knowledge, and she is free to use any algorithms or models. Even if the model chosen initially is inaccurate because of masking, the interactive test progressively reveals the assignments (of the first $t-1$ subjects at step $t$) to the analyst, so that she can improve her understanding of the data and update the model or heuristic for estimating the treatment assignments at any step.

We call our proposed procedure the i-bet test. Our contribution is to provide a "game-theoretic" principle for the causal hypothesis testing problem, and demonstrate a new class of interactive multi-step algorithms that, by masking some of the data and progressively revealing it to the scientist, can combine the strengths of (automated) statistical modeling and (human-guided) scientific knowledge, in order to reject the global null while not suffering from any p-hacking or data-dredging concerns despite a great deal of flexibility provided to the scientist.

**Directly related work.** The game-theoretic principles of testing by betting stems from the books by Shafer and Vovk (2019, 2005) as elucidated in a recent paper by Shafer (2020). Recently, these

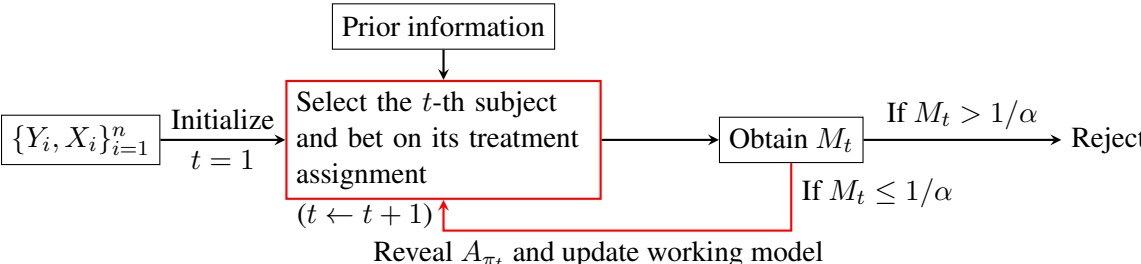

Figure 1: Schematics of the i-bet test. At each step, a human analyst can freely explore and update models to guide the selection of the $t$-th subject and its bet (as the red box shows).

ideas have been successfully applied to election auditing (Waudby-Smith et al., 2021), and for constructing concentration bounds (Waudby-Smith and Ramdas, 2020). Our work connects the betting perspective with the causal null hypothesis, introducing the various advantages as outlined in the abstract. The advantage of optional continuation of experiments (equivalently, optional stopping) has been highlighted in particular by Grünwald et al. (2019) and Howard et al. (2021).

The idea of interactive testing was recently proposed by Lei and Fithian (2018) and Lei et al. (2020), in the context of multiple testing problems to control FDR (the false discovery rate), followed by several works for other error metrics in multiple testing. Our interactive test for two-sample comparison relates most with the work of controlling the global type-I error (Duan et al., 2020), where the individual null hypothesis is zero effect for each subject, and the global null corresponds to the null of no treatment effect as null hypothesis (1). Previous development of the interactive tests typically focuses on generic multiple testing problems, which operate directly on multiple $p$-values; in other words, the units of inference were $p$-values (for different hypotheses) rather than data points (to test a single hypothesis). Here, interactive testing is directly applied to the observed data, expanding the potential of interactive tests.

For the related problem of two-sample testing, Lhéritier and Cazals (2018) developed the same idea of constructing the test statistics as a nonnegative martingale. One can view our paper as an extension of their work to causal inference settings, developing the core idea further along methodological, theoretical and practical fronts (for example, extending to sampling without replacement in order to handle different types of randomization beyond independent Bernoulli assignment). We emphasize the flexibility allowed to a human analyst to utilize arbitrary contextual knowledge prior to and during the test, with or without the aid of probabilistic modeling. We also develop some extensions in Appendix E. Other related work on testing weaker null hypotheses is in Appendix F.

**Outline.** The rest of the paper is organized as follows. In Section 2, we describe the i-bet test in detail, followed by numerical experiments to demonstrate its advantage over standard methods in Section 3. Section 4 concludes the paper with a discussion on the potential of interactive rank tests. Extensions to various settings, such as paired data, are deferred to Appendix E.

## 2. An interactive rank test with covariates (i-bet)

To account for covariates through a flexible algorithm that involves human interaction, we propose the i-bet test. In short, the analyst decides the ordering of subjects $\{\pi_j\}_{j=1}^n$ and the bets $\{w_j\}_{j=1}^n$

progressively: at step $t$, she selects the $t$-th subject from the to-be-ordered subjects $[n]\backslash\{\pi_j\}_{j=1}^{t-1}$ and decides the bet $w_t$, based on an increasing amount of data information starting from the assignments $\{A_i\}_{i=1}^n$ masked and then gradually revealed. Note that the bet can be unreliable, and in turn hurt the power, at the first few steps if all the assignments are masked. Thus, we reveal the complete data for a random subset of subjects at $t = 0$, denoted as set $\mathcal{I}_0$ (10% of all subjects for example). At each iteration, we select subject $\pi_t$ and expand the set $\mathcal{I}_t = \mathcal{I}_{t-1} \cup \{\pi_t\}$ whose complete data is then revealed.

Mathematically, the data information available to the analyst at the end of step $t$ is denoted by the filtration:

$$\mathcal{F}_t = \sigma\left(\{Y_i, X_i\}_{i=1}^n \cup \{A_i\}_{i \in \mathcal{I}_t}\right). \tag{7}$$

The choice of $\pi_t$ and $w_t$ are predictable (measurable) with respect to $\mathcal{F}_{t-1}$, while the analyst is allowed to explore and choose arbitrary models or heuristics to form the ordering and get the bets. After each iteration of selecting $\pi_t \in [n]\backslash\mathcal{I}_{t-1}$ and choosing $w_t \in [-\frac{1}{1-\mu_{\pi_t}}, \frac{1}{\mu_{\pi_t}}]$, the test calculates

$$M_t = \prod_{j=1}^t \left[1 + w_j \cdot (A_{\pi_j} - \mu_{\pi_j})\right], \tag{8}$$

and the iteration stops once $M_t$ reaches the boundary $1/\alpha$, or all the subjects in $[n]\backslash\mathcal{I}_0$ are ordered. We summarize the i-bet test in Algorithm 2.

**Input:** Outcomes, assignments, covariates $\{Y_i, A_i, X_i\}_{i=1}^n$, target type-I error $\alpha$, holdout ratio $\gamma$;
**Procedure:** 1. Random select a set $\mathcal{I}_0 \in [n]$ with size $\gamma \cdot n$;
**for** $t = 1, \cdots, |[n]\backslash\mathcal{I}_0|$ **do**
  2. Using $\mathcal{F}_{t-1}$, pick any $\pi_t \in [n]\backslash\mathcal{I}_{t-1}$ and obtain an arbitrary bet $w_t \in [-\frac{1}{1-\mu_{\pi_t}}, \frac{1}{\mu_{\pi_t}}]$;
  3. Reveal $A_{\pi_t}$ and update $\mathcal{I}_t$ and $\mathcal{F}_t$;
  **if** $\prod_{j=1}^t \left[1 + w_j \cdot (A_{\pi_j} - \mu_{\pi_j})\right] > 1/\alpha$ **then**
    | Reject the null and stop;
**end**
  **Algorithm 2:** Framework for the interactive rank test (i-bet test)

## 2.1. Important remarks

**Remark 3** *We defined the problem as testing the global null* (1) *of no treatment effect at a predefined level $\alpha$. Instead, we could ask the test to output a p-value for the global null, or even better, to output an anytime-valid p-value, which is a sequence of p-values $\{p_t\}_{t=1}^n$ such that for an arbitrary stopping time $\tau$, $p_\tau$ is also a valid p-value (its distribution is stochastically larger than uniform if the null is true). Luckily, this is easy: $p_t = \inf_{s \leq t} 1/M_s$ fits the bill, once again due to Ville's inequality. Further, being a nonnegative martingale, the optional stopping theorem implies that the wealth process at any stopping time has expectation at most one under the null; this makes the wealth process an* e-process *(Ramdas et al., 2021; Grünwald et al., 2019; Shafer, 2020). The relationship between these objects is detailed in Ramdas et al. (2020).*

**Remark 4** *The anytime-validity discussed above implies that the experiment can be extended to a larger size if the smaller size did not provide sufficient evidence (meaning the wealth did not exceed $1/\alpha$), as discussed in Grünwald et al. (2019) and Howard et al. (2020). It is indeed a remarkable*

*property that if the null cannot be rejected using the current dataset, we can just continue experimentation: randomly assign $l$ more people to treatment and control, reveal their covariates and outcomes, and continue the betting on the new $l$ subjects starting with the wealth $M_n$, extending the original ranking as if we had all $l + n$ subjects from the start. This optional continuation does not require adjustment for multiple testing before or after collecting new samples, because the wealth continues to be a nonnegative martingale under the null, the $p$-value is anytime-valid, and the probability the wealth ever exceeds $1/\alpha$ is at most $\alpha$. In game-theoretic terms, no amount of betting can make us significantly rich in a fair game (characterized by the martingale property under the null) — even if we first chose to play $n$ rounds, and then later added $l$ more rounds.*

**Remark 5** *The i-bet test can be extended to a completely randomized experiment, where the number of treatments is fixed and known as $m$ at the beginning, and $m$ subjects are randomly chosen to be treated. In such a case, the revealed information additionally has the sum of treatment assignments*

$$\mathcal{F}_t = \sigma \left( \{Y_i, X_i\}_{i=1}^n \cup \{A_i\}_{i \in \mathcal{I}_t} \cup \{\sum_{i=1}^n A_i\} \right).$$

*We would then construct the nonnegative martingale as $M_t = \prod_{j=1}^t \left[ 1 + w_j \cdot (A_{\pi_j} - \mu_j) \right]$, where $w_t \in [-\frac{1}{1-\mu_t}, \frac{1}{\mu_t}]$ and $\mu_t = \left( m - \sum_{i \in \mathcal{I}_{t-1}} A_i \right) / (n - |\mathcal{I}_{t-1}|)$ is the expected treatment assignment given the revealed information in $\mathcal{F}_{t-1}$.*

**Remark 6** *Despite high flexibility in choosing the weight $w_t$ and the order $\pi_t$, good choices can increase power. The choice of the ordering $\pi_t$ affects the test power, when taken together with the choice of weight $w_t$. So let us first note that a desirable weight $w_t$ should ideally have the same sign as $A_{\pi_t}$; this would allow $M_t$ to increase to sooner reach the rejection threshold $1/\alpha$. Therefore, we recommend practitioners to order upfront the subjects for which they (or the algorithm acting on their behalf) are most confident about their treatment assignment. As an example, we provide an automatic approach to choose weight $w_t$ and order $\pi_t$ in Section 2.2.*

**Theorem 7** *As long as an analyst explores and updates working models at any step $t$ using only the information in $\mathcal{F}_t$, the i-bet test controls type-I error for null hypothesis (1) under assumptions (i),(ii) of randomized experiments. In fact, the error control holds conditionally on $\{X_i, Y_i\}_{i=1}^n$.*

Although more information is revealed to the analyst after each step, the error control is valid, because under the null, the increment $A_{\pi_t}$ for testing is independent of the revealed information:

$$\mathbb{E} \left( A_{\pi_t} \mid \mathcal{F}_{t-1} \right) = \mu_{\pi_t}. \tag{9}$$

The complete proof is in Appendix A.

The i-bet allows the analyst to incorporate covariates and various types of domain knowledge for ordering and choosing weights. However, manually picking $\pi_t$ at every step could be tedious. The analyst can instead design an automated algorithm for choosing $\pi_t$ and $w_t$, such as the example we provide in the next section, and still keeps the flexibility to modify it at any step.

## 2.2. A concrete, automated, instantiation of i-bet

We can infer the treatment assignments by exploring various models to fit the (partial) data. An example is to model the outcome as a mixture of the distributions for treatment and control groups:

$$Y_i \sim \begin{cases} N(\mu_i^1, 1), & \text{when } A_i = 1 \\ N(\mu_i^0, 1), & \text{when } A_i = 0 \end{cases} \quad \text{with } \mu_i^j = \theta_j(X_i) \text{ for } j = 0, 1, \tag{10}$$

where $\theta_j$ could be linear functions of the covariates and their second-order interaction terms. The masked treatment assignments can be viewed as missing values, and by the EM algorithm (details in Appendix B), we get an estimated posterior probability of receiving the treatment for each subject. The estimated probability of receiving treatment, denoted as $\widehat{q}_i$, provides an estimation of the assignment and an approach to select $\pi_t$. Recall that we hope to order upfront the subject whose estimated assignment we are most confident, which can be measured by $|\widehat{q}_i - 0.5|$, so we could select $\pi_t = \arg\max_{i \in [n] \setminus \{\pi_j\}_{j=1}^{t-1}} \{|\widehat{q}_i - 0.5|\}$. For the chosen subject, we bet on the treatment assignment by $w_j = 0.8(2\widehat{A}_{\pi_j} - 1)$, where the estimated assignment $\widehat{A}_{\pi_j} := \mathbb{1}\{\widehat{q}_{\pi_j} > 0.5\}$ is a function of the estimated probability of receiving treatment. We summarize this automated procedure in Algorithm 3.

**Input:** Outcomes, assignments, covariates $\{Y_i, A_i, X_i\}_{i=1}^n$, target type-I error $\alpha$, holdout ratio $\gamma$;
**Procedure:** 1. Random select a set $\mathcal{I}_0 \in [n]$ with size $\gamma \cdot n$;
**for** $t = 1, \cdots, |[n] \setminus \mathcal{I}_0|$ **do**
    2. Estimate $\widehat{q}_i$ for subjects in $[n] \setminus \mathcal{I}_{t-1}$;
    3. Choose $\pi_t = \arg\max_{i \in [n] \setminus \mathcal{I}_{t-1}} \{|\widehat{q}_i - 0.5|\}$;
    4. Reveal $A_{\pi_t}$ and update $\mathcal{I}_t$ and $\mathcal{F}_t$;
    **if** $\prod_{j=1}^t \left[1 + 0.8(2\mathbb{1}\{\widehat{q}_{\pi_j} > 0.5\} - 1) \cdot (A_{\pi_j} - 1/2)\right] > 1/\alpha$ **then**
        | Reject the null and stop;
**end**

**Algorithm 3:** An automated implementation of the i-bet test

By design, $w_j \cdot (A_{\pi_j} - 1/2)$ is $+0.4$ if the estimated assignment is consistent with the truth; and $-0.4$ otherwise [2]. Ideally, when the null is false, we could guess most assignments correctly and order them upfront, leading to a larger $M_t$ that could exceed the boundary $1/\alpha$.

As the test proceeds and more actual assignments get revealed for interaction, we refit the above model and update the estimation of posterior probabilities for every $\lfloor \frac{n}{5} \rfloor$ steps (say). Keep in mind that the validity of the error control does not require model (10) to be correct. The analyst can choose other models such as logistic regression for $\theta_j$ if the revealed data or prior knowledge suggests so.

## 3. Numerical experiments

Though the primary contribution of our paper is the construction and derivation of a conceptually interesting test (or framework, since there is significant flexibility left to the analyst) for the global causal null, we attempt below to convince the reader that the flexibility afforded by our interactive setup suffices to deliver high power. We view the simulations with i-bet as a thought experiment: the reader must imagine that we perhaps chose a poor model for all methods at the start. Using a

---

2. We could design bets $w_j$ such that $w_j \cdot (A_{\pi_j} - 1/2)$ has larger contrast (e.g. $\pm 1$), but over-betting would hurt power.

poor (uninformative or barely better than chance) model, our bet $w_j$ would possibly have a random sign for early subjects, and our wealth may fluctuate up and down, rather than increase reasonably steadily. The multi-step i-bet test can have higher power than other single-shot tests because, in the midst of this testing, the analyst can observe the poor start, explore and evaluate various models using the complete data for completed bets and the masked-assignment data for every other point, and try to find a better model (details in the paragraph of "Illustration of adaptive modeling").

**Simulation setup.** To evaluate the performance of the automated algorithm, we simulate 500 subjects ($n = 500$). Suppose each subject is recorded with two binary attributes (e.g., female/male and senior/junior) and one continuous attribute (e.g., body weight), all of which are denoted as a vector $X_i = (X_i(1), X_i(2), X_i(3)) \in \{0,1\}^2 \times \mathbb{R}$. Among $n$ subjects, the binary attributes are marginally balanced, and the subpopulation with $X_i(1) = 1$ and $X_i(2) = 1$ is of size $n_0$ (see Table 1), where we set $n_0 = 30$. The continuous attribute is independent of the binary ones and follows the distribution of a standard Gaussian.

Table 1: Size of the subpopulation in terms of two binary attributes.

|  | $X_i(1) = 0$ | $X_i(1) = 1$ | Totals |
|---|---|---|---|
| $X_i(2) = 0$ | $n_0$ | $n/2 - n_0$ | $n/2$ |
| $X_i(2) = 1$ | $n/2 - n_0$ | $n_0$ | $n/2$ |
| Totals | $n/2$ | $n/2$ | $n$ |

The outcomes are simulated as a function of the covariates $X_i$ and the treatment assignment $A_i$ following the generating model (2), where we vary the functions for the treatment effect $\Delta$ and the control outcome $f$ to evaluate the performance of the i-bet test. Recall that earlier, we used model (2) as a working model, which is not required to be correctly specified. Here, we generate data from such a model in simulation to provide various types of underlying truth for a clear evaluation of the considered methods[3].

**Alternative tests for comparison.** In addition to the CovAdj Wilcoxon test, we compare the i-bet test with a semi-parametric test derived from the literature of estimating conditional average treatment effect (CATE), which we refer to as the linear-CATE-test. Here, the nonparametric testing problem is transformed into testing a parameter, potentially considering a less stringent null. Specifically, null hypothesis (1) implies that

$$\text{if } \mathbb{E}(Y_i \mid A_i = 1, X_i) - \mathbb{E}(Y_i \mid A_i = 0, X_i) = X_i^T \psi^*, \text{ then } \psi^* = \mathbf{0}. \tag{11}$$

Assume that the outcome difference is a linear function of covariates $X_i$, the method for CATE provides an asymptotic confidence interval for $\psi^*$, and the null is rejected if the confidence interval does not include zero (see Appendix C for an explicit form of the test). Note that the test has valid error control even if the outcome difference is not linearly correlated with $X_i$, in which case, however, the power would be low.

The presented methods (the CovAdj Wilcoxon test, the linear-CATE-test, and the automated algorithm of the i-bet test) all involve some working model of the outcomes, but the extent of flexibility varies. The linear-CATE-test requires us to specify the parametric model before looking

---

3. R code to reproduce all plots in the paper is available in https://github.com/duanby/interactive-rank. When implementing algorithm 3, we choose the holdout ratio $\gamma = 0.1$ by default.

at the data; the CovAdj Wilcoxon test allows model exploration given partial data $\{Y_i, X_i\}_{i=1}^n$ before testing; and the i-bet test further permits the analyst to interactively change the model as the test proceeds and more assignments $A_i$ become available for modeling.

**Test performances when the default model is a good fit.** Consider outcomes from the generating model (2) with the treatment effect $\Delta$ and the control outcome $f$ specified as:

$$\Delta(X_i) = S_\Delta[X_i(1) \cdot X_i(2) + X_i(3)], \tag{12}$$

$$f(X_i) = 5[X_i(1) + X_i(2) + X_i(3)], \tag{13}$$

where $S_\Delta$ encodes the signal strength of the effect. Intuitively, all subjects have some Gaussian-distributed effect correlated with $X(3)$ and the subjects with $X(1) = 1$ and $X(2) = 1$ additionally have a constant positive effect. In such a setting, all the methods with their working models specified as linear functions should fit the data well.

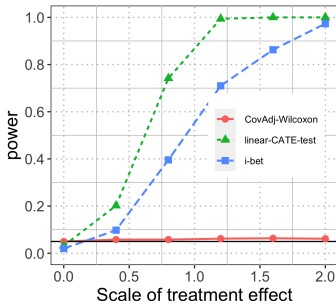

Figure 2: Power of the i-bet test compared with the standard tests when varying the scale of the treatment effect, which is defined in (12). The linear model used in all the tests is a good fit for the underlying truth, and the linear-CATE-test (21) has higher power. In plots of this section, the power is averaged over 500 repetitions (estimated by the proportion of repeated experiments where the null gets rejected), and the error bar is omitted because its length is usually less than 0.02.

For two-sided heterogeneous treatment effects, the CovAdj Wilcoxon test has low power because the positive effects cancel out with the negative effects in the sum statistics (4), while the linear-CATE-test and the i-bet test can accumulate the effect of both signs. The linear-CATE-test has higher power as it targets the specific alternative of nonzero parameters in the linear model (11), although the i-bet test also achieves reasonable power in Figure 2. Note that the three methods we compare (CovAdj Wilcoxon test, linear-CATE, i-bet test) all have valid type-I error control for the same global null $H_0$, while these methods implicitly target alternatives in different directions. Recall that we do not make any assumptions on the distribution of non-nulls — that is, if some people do respond to treatment, no assumption is made on how they respond — or how informative the covariates are. It is well known that in such nonparametric settings, there is no universally most powerful test; for example, Janssen (2000) discusses this phenomenon when testing goodness of fit. We demonstrate next that i-bet could have higher power when the initial working model may be incorrect, among other situations.

**Illustrations of adaptive modeling.** One advantage of the interactive test is that it allows exploration and adaptation of the working model using the revealed data. Here, we present an example where model (10) might not fit the data well; the reader must imagine that we do not suspect this at the start, so we begin by utilizing it anyway. However, suppose the poor guidance provided by the incorrect model results in the algorithm making mistakes guessing the assignments at the first few steps itself (even though we are ordering them from most to least confident!). By a mistake, we mean that our bet $w_j$ had the wrong sign, and we lose some wealth. The multi-step i-bet test can have higher power than other single-shot tests because, in the midst of this testing, the analyst can observe the poor start, explore and evaluate various models, and find a reasonably good fit using all the available revealed data. (Practitioners using the one-shot methods may change their model and thus their test statistic after observing its performance on the full data, for example whether it rejects or not, but this technically invalidates their type-I error guarantee.)

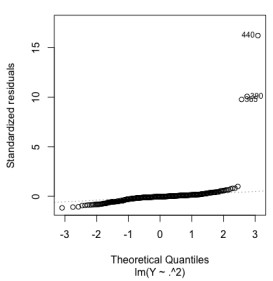

(a) Eg: diagnose a misfit via QQ-plot for the original linear model before testing.

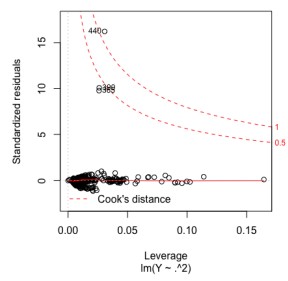

(b) Eg: diagnose a misfit via Cook's distance for the original linear model before testing.

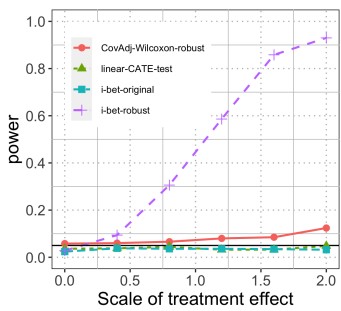

(c) Power under skewed control outcome (14).

Figure 3: Before ordering and testing, the analyst is allowed to explore and examine different working models using the revealed data $\{Y_i, X_i\}_{i=1}^n$. In the example with skewed control outcome, the QQ-plot and Cook's distance of the regular linear regression suggest outliers in the outcomes. The analyst can instead choose the robust linear regression, and the power is higher than that using the default model. For fair comparison, the CovAdj Wilcoxon test (4) is also implemented with robust linear regression.

Suppose the control outcome is nonlinearly correlated with the attributes by specifying function $f$ in the generating model (2) as

$$f(X_i) = 2\exp\{-2X_i(3)\}\mathbb{1}(X_i(3) < -2), \tag{14}$$

where the distribution of potential control outcomes is skewed (treatment effect $\Delta$ is the same as before in (12)). When we fit the default working model (10) with linear functions (along with a few revealed treatment assignments), the QQ-plot and Cook's distance indicate a poor fit because of possible outliers in the outcomes (Figures 3(a) and 3(b)). An easy fix is to use robust linear regression (Huber, 2004), which leads to significant power improvement compared with the default algorithm (see Figure 3(c)). (In practice, we recommend using robust regression from the very beginning anyway since it keeps good power when the working model is correct while it improves

power when the control outcome has a skewed distribution. The robust regression is also observed to improve power under heavy-tailed noise (see Appendix D.1). However, for the purpose of this illustrative example, presume that we switch to it after observing a few early mistakes.) Another example of model exploration considers treatment effect as a quadratic function of the covariates, and a robust regression with the quadratic term can perform better (see Appendix D.2).

**Real-data application to the effect of screening ASB on reduction of low birth weight.** We use the data in Gehani et al. (2021), which investigates whether antenatal screening of asymptomatic bacteriuria (ASB) can reduce low birth weight. The study collected 240 participants and randomly selected 120 of them into the treated group (and the rest in the control group). The treated group additionally screened for ASB with a novel rapid test; and the control group did not receive such a test. The outcome $Y_i$ is the birth weight. The covariates $X_i$ include trimester at the time of enrolment, gestation age, and the number of previous pregnancies. We use a robust linear model in the i-bet test, as recommended in Figure 3. The final wealth $M_n$ after betting for all participants is $10^{15}$ (see the path of log-wealth in Appendix D.3), and the $p$-value is $10^{-16}$ (it is the inverse of the maximum wealth, not final wealth). As a comparison, the original paper evaluates the effect by a binary indicator for low birth weight, and they also detect statistically significant difference between treated and control group.

To summarize, the i-bet test has valid error control without any parametric assumption on the outcomes and allows exploration of working models so that the algorithm can adapt to different underlying data distributions. *In practice, the working model can also be changed in the middle of the testing procedure, for example, if it fits the data worse as more treatment assignments get revealed.* The flexibility of interactive data-dependent model design with the freedom of adjustment on the fly makes the i-bet test with parametric working models practical and promising.

## 4. Summary

For randomized trials, we have proposed the i-bet test, which takes the perspective of betting and incorporates the recent idea of allowing human interaction via the procedure of "masking" and "unmasking". The interactive tests encourage the analyst to explore various working models before and during the testing procedure, so that the test can integrate the observed data information with prior knowledge of various types and even a human's subjective belief in a highly flexible manner.

Due to space, we have only discussed in depth the setting of two-sample comparisons with unpaired data, and our test can be extended to various problem settings: two/multi-sample comparison with/without block structure, and a dynamic setting with subjects or mini-batches of subjects arrive sequentially (see Appendix E). The current i-bet test can have unstable results when implemented on real datasets with binary outcomes. As a future direction, we hope to polish the models for deciding the ordering $\pi_t$ and bet $w_t$ in related settings, such as with binary outcomes, or with the proportion of treated subjects being small.

We remark that no test, interactive or otherwise, can be run twice from scratch (with a tweak made the second time to boost power) after the entire data has been examined; this amounts to $p$-hacking. Our interactive tests—that can be continued with additional experimentation—are one step towards enabling experts (scientists and statisticians) to work together with statistical models and machine learning algorithms in order to discover scientific insights with rigorous guarantees.

## Acknowledgments

We thank the anonymous reviewers for their helpful suggestions. AR acknowledges support from NSF DMS 1916320, and NSF CAREER 1945266. LW acknowledges support from NSF DMS 1713003. This work used the Extreme Science and Engineering Discovery Environment (XSEDE) (Towns et al., 2014), which is supported by National Science Foundation grant number ACI-1548562. Specifically, it used the Bridges system Nystrom et al. (2015), which is supported by NSF award number ACI-1445606, at the Pittsburgh Supercomputing Center (PSC).

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

## Appendix A. Proof of Theorem 7

**Proof** We argue that the product $\{M_t\}_{t=1}^{|[n]\setminus\mathcal{I}_0|}$ is a nonnegative martingale with respect to the filtration $\{\mathcal{F}_{t-1}\}_{t=1}^{|[n]\setminus\mathcal{I}_0|}$. First, the product $M_t$ is measurable with respect to $\mathcal{F}_{t-1}$, because $M_t = \prod_{j=1}^{t} [1 + w_j \cdot (A_{\pi_j} - \mu_{\pi_j})]$, where the $t$-th selected subject $\pi_t$ and its bet $w_t$ are all $\mathcal{F}_{t-1}$-measurable.

Second, we show that $\mathbb{E}(M_t \mid \mathcal{F}_{t-1}) = M_{t-1}$. Note that $M_{t-1}$ is fixed given $\mathcal{F}_{t-1}$, so $\mathbb{E}(M_t \mid \mathcal{F}_{t-1}) = M_{t-1} \cdot \mathbb{E}[1 + w_t \cdot (A_{\pi_t} - \mu_{\pi_t}) \mid \mathcal{F}_{t-1}]$, and $\mathbb{E}(M_t \mid \mathcal{F}_{t-1}) = M_{t-1}$ holds when $\mathbb{E}(w_t \cdot (A_{\pi_t} - \mu_{\pi_t}) \mid \mathcal{F}_{t-1}) = 0$, which is implied when

$$\mathbb{E}(A_{\pi_t} \mid \mathcal{F}_{t-1}, w_t, \pi_t) = \mu_{\pi_t}. \tag{15}$$

The above can be verified because

$$\mathbb{E}(A_{\pi_t} \mid \mathcal{F}_{t-1}, w_t, \pi_t) \overset{(a)}{=} \mathbb{E}(A_{\pi_t} \mid \mathcal{F}_{t-1}, \pi_t)$$

$$= \sum_{j=1}^{n} \mathbb{E}(A_j \mid \mathcal{F}_{t-1}, \pi_t = j) \mathbb{1}(\pi_t = j \mid \mathcal{F}_{t-1}, \pi_t) \overset{(b)}{=} \sum_{j=1}^{n} \mathbb{E}(A_j \mid \mathcal{F}_{t-1}) \mathbb{1}(\pi_t = j \mid \mathcal{F}_{t-1}, \pi_t)$$

$$\overset{(c)}{=} \sum_{j=1}^{n} \mathbb{E}(A_j \mid X_j) \mathbb{1}(\pi_t = j \mid \mathcal{F}_{t-1}, \pi_t) = \sum_{j=1}^{n} \mu_j \mathbb{1}(\pi_t = j \mid \mathcal{F}_{t-1}, \pi_t) = \mu_{\pi_t},$$

where $(a)$ holds because $w_t$ is $\mathcal{F}_{t-1}$-measurable, and $(b)$ is because $\pi_t$ is $\mathcal{F}_{t-1}$-measurable, and $(c)$ stems from the fact that $A_j$ is independent of all the outcomes and other treatment assignments and covariates under the global null. Also, the fact that $A_{\pi_j} \in \{0, 1\}$ and $w_j \in [-2, 2]$ ensures $M_t$ to be nonnegative at any time $t$. Thus, we conclude that $\{M_t\}_{t=1}^{|[n]\setminus\mathcal{I}_0|}$ is a nonnegative martingale. The error control follows by Ville's inequality (6). ∎

## Appendix B. Estimation of the posterior probability of receiving treatment

Under working model (10), we view the treatment assignments of to-be-ordered subjects as hidden variables and apply the EM algorithm. At step $t$, the hidden variables are $A_i$ for subjects $i \notin \{\pi_j\}_{i=1}^{t-1}$. And the rest of the complete data $\{Y_i, A_i, X_i\}_{i=1}^{n}$ is the observed data, denoted by $\sigma$-field $\mathcal{F}_{t-1}$ as defined in (7). In the working model (10), the log-likelihood of $\{Y_i, A_i, X_i\}_{i=1}^{n}$ is

$$l\left(\{Y_i, A_i, X_i\}_{i=1}^{n}\right) = \sum_{i \in [n]} [A_i \log \phi\left(Y_i - \theta_1(X_i)\right) + (1 - A_i) \log \phi\left(Y_i - \theta_0(X_i)\right) + g(X_i)],$$

where $\phi(\cdot)$ is the density of standard Gaussian and $g(\cdot)$ denotes the density of the covariates. In the E-step, we update the hidden variable $A_i$ for $i \notin \{\pi_j\}_{i=1}^{t-1}$ as

$$A_i^{\text{new}} = \mathbb{E}(A_i \mid \mathcal{F}_{t-1}) = \frac{\phi\left(Y_i - \theta_1(X_i)\right)}{\phi\left(Y_i - \theta_1(X_i)\right) + \phi\left(Y_i - \theta_0(X_i)\right)}.$$

In the M-step, we update the (parametric) functions $\theta_0$ and $\theta_1$ as

$$\theta_0^{\text{new}} = \arg\max l\left(\{Y_i, A_i, X_i\}\right) = \arg\min \sum_{i \in [n]} (1 - A_i)(Y_i - \theta_0(X_i))^2,$$

$$\theta_1^{\text{new}} = \arg\max l\left(\{Y_i, A_i, X_i\}\right) = \arg\min \sum_{i \in [n]} A_i(Y_i - \theta_1(X_i))^2,$$

which are least square regressions with weights. The posterior probability of receiving treatment is estimated as $\mathbb{E}(A_i \mid \mathcal{F}_{t-1})$ for $i \notin \{\pi_j\}_{i=1}^{t-1}$.

## Appendix C. The linear-CATE-test

We first describe the general framework of CATE without specifying the working model (see Vansteelandt and Joffe (2014) for a review). Suppose $\psi^*$ is a vector of parameters, and a pre-defined function $h$ satisfies $h(\psi^*, x) = 0$ if $\psi^* = 0$, for which a standard choice is a linear function of the covariates, $h(\psi^*, x) = x^T \psi^*$. One first posits that the difference in conditional expectations satisfies

$$\mathbb{E}(Y_i \mid X_i, A_i = 1) - \mathbb{E}(Y_i \mid X_i, A_i = 0) = h(\psi^*, X_i). \tag{16}$$

Thus, a valid test for null hypothesis (1) can be developed by testing $\psi^* = 0$. Note that the test is model-free (regardless of the correctness of $h$) since $\psi^* = 0$ is implied by null hypothesis (1) for any function $h$ specified as above. The inference on $\psi^*$ uses an observation that for any function $g$ of the covariates and the assignment, we have

$$\mathbb{E}\{[g(X_i, A_i) - \mathbb{E}(g(X_i, A_i) \mid X_i)] \cdot [Y_i - \mathbb{E}(Y_i \mid X_i, A_i)]\} = 0, \tag{17}$$

where $\mathbb{E}(Y_i \mid X_i, A_i) = A_i \cdot h(\psi^*, X_i) + \mathbb{E}(Y_i \mid X_i, A_i = 0)$ because of (16). To estimate $\psi^*$, we need to specify functions $h$ and $g$, and estimate $\mathbb{E}(g(X_i, A_i) \mid X_i)$ and $\mathbb{E}(Y_i \mid X_i, A_i = 0)$. Notice that in a randomized experiment, $\mathbb{E}(g(X_i, A_i) \mid X_i)$ is known given $g$, which guarantees that equation (17) holds regardless of whether $\mathbb{E}(Y_i \mid X_i, A_i = 0)$ is correctly specified (double robustness). In the following, we choose functions $h$, $g$ and estimate $\mathbb{E}(Y_i \mid X_i, A_i = 0)$ without being concerned about the validity of equation (17). After getting an estimator of $\psi^*$, we present the test for $\psi^* = 0$ in the end.

For fair comparison with the i-bet test that uses linear model by default, we set $h$ to be a linear function of the covariates and their second-order interaction terms. Let $X_i'$ be the vector of covariates $X_i$ and the interaction terms, then $h = (X_i')^T \psi^*$. In such as case, a good choice of function $g$ is $X_i' \cdot A_i$ (Vansteelandt and Joffe, 2014). Because other methods in our comparison use linear models by default, we estimate $\mathbb{E}(Y_i \mid X_i, A_i = 0)$ by a linear model of $X_i'$, denoted as $(X_i')^T \widehat{\beta}$ (note that $\widehat{\beta}$ can be learned by regressing $Y_i$ on $X_i$ without involving $A_i$ since under the null, $\mathbb{E}(Y_i \mid X_i, A_i = 0) = \mathbb{E}(Y_i \mid X_i, A_i = 1) = \mathbb{E}(Y_i \mid X_i)$). With the above choices, equation (17) can be written as

$$\mathbb{E}\left[\underbrace{(A_i - 1/2)(Y_i - (X_i')^T \widehat{\beta}) X_i'}_{b_i}\right] = \mathbb{E}\left[\underbrace{(X_i'^T A_i(A_i - 1/2) X_i')}_{B_i} \psi^*\right], \tag{18}$$

which is denoted as $\mathbb{E}(b_i) = \mathbb{E}(B_i)\psi^*$ for simplicity. Let $\mathbb{P}_n b$ be the sample average of $\{b_i\}_{i=1}^n$ and $\mathbb{P}_n B$ be the sample average of $\{B_i\}_{i=1}^n$. A consistent estimator of $\psi^*$ is

$$\widehat{\psi} = (\mathbb{P}_n B)^{-1} \mathbb{P}_n b \tag{19}$$

$$= \left(\frac{1}{n}\sum_{j=1}^n X_j'^T A_j(A_j - 1/2) X_j'\right)^{-1}\left(\frac{1}{n}\sum_{i=1}^n (A_i - 1/2)(Y_i - (X_i')^T \widehat{\beta}) X_i'\right).$$

The test statistic is proposed based on $\psi^*$ and its variance estimator. Notice that the asymptotic variance of $\widehat{\psi}$ is $B^{-1}\mathrm{Var}(b)(B^{-1})^T$ conditional on $\{X_i, A_i\}_{i=1}^n$, for which a consistent estimator is

$$\widehat{\mathrm{Var}}(\psi) = (\mathbb{P}_n B)^{-1}\,\widehat{\mathrm{Var}}(b)\left[(\mathbb{P}_n B)^{-1}\right]^T, \tag{20}$$

where $\widehat{\mathrm{Var}}(b)$ denotes the sample covariance of $\{b_i\}_{i=1}^n$. Thus, the test statistic is proposed as

$$S = \widehat{\psi}^T[\widehat{\mathrm{Var}}(\psi)]^{-1}\widehat{\psi} = (\mathbb{P}_n b)^T[\widehat{\mathrm{Var}}(b)]^{-1}(\mathbb{P}_n b).$$

The limiting distribution of $S$ under the null is $\chi_p^2$ where $p$ is the dimension of $X_i'$, because $\mathbb{E}[b_i] = \mathbb{E}(B_i)\psi^* = 0$ under the null. The linear-CATE-test rejects the null if

$$(\mathbb{P}_n b)^T[\widehat{\mathrm{Var}}(b)]^{-1}(\mathbb{P}_n b) > \chi_p^2(1-\alpha), \tag{21}$$

where $b_i$ is defined in (18); and $\mathbb{P}_n$ and $\widehat{\mathrm{Var}}$ denotes sample average and sample covariance matrix; and $\chi_p^2(1-\alpha)$ is the $1-\alpha$ quantile of a chi-squared distribution with $p$ degrees of freedom.

## Appendix D. Experiments for the i-bet test

### D.1. Heavy-tailed noise

In the automated algorithm of i-bet test, we recommend using the robust regression because it is less sensitive to skewed control outcomes, as shown by Figure 3(c). Here, we show that the robust regression also makes the i-bet test more robust to heavy-tailed noise (see Figure 4).

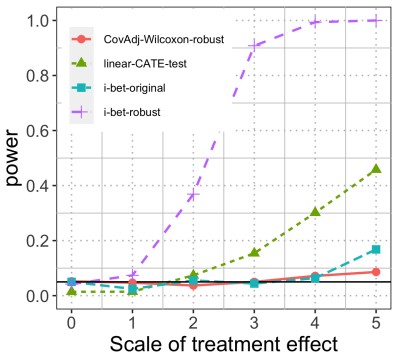

Figure 4: Power of the i-bet test using regular linear regression and robust linear regression compared with standard methods. The outcome simulates from (2), where the function of treatment effect $\Delta$ and the function of control outcome $f$ are linear as defined in (12) and (13). Instead of Gaussian noise in Section 3, the noise $U_i$ is now Cauchy distributed. The i-bet test with robust linear regression has higher power than that using regular linear regression under heavy-tailed noise. For a fair comparison, the CovAdj Wilcoxon test is also implemented with robust linear regression.

### D.2. Quadratic treatment effect

Another example of adaptive modeling considers the treatment effect as a quadratic function of the covariates, by specifying the function $\Delta$ in the generating model (2) as

$$\Delta(X_i) = S_\Delta \left[ \frac{3}{5} \left( X_i^2(3) - 1 \right) \right]. \tag{22}$$

The control outcome is linearly correlated with the attributes as defined in (13). We observe that with the robust linear regression, the residuals have a nonlinear trend (see Figure 5(a)), indicating that the linear functions of covariates might not be accurate. If we add a quadratic term of $X_i^2(3)$ in the robust regression, the trend in residuals is less obvious, and the model fits better (see Figure 5(b)). As a result, the power is higher than the test using robust linear regression (see Figure 5(c)).

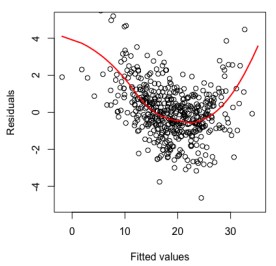 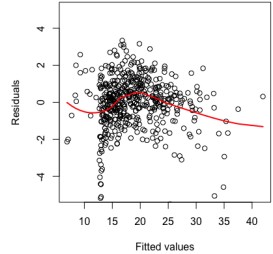 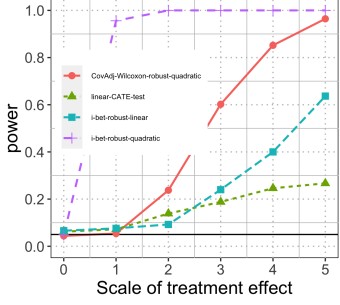

(a) Residual plot when using the robust linear regression.

(b) Residual plot when applying regression with a quadratic term.

(c) Power when the treatment effect is a nonlinear function of the covariates.

Figure 5: A second illustration of model exploration when the treatment effect is nonlinearly correlated with the attributes. The residuals show a quadratic pattern when using robust linear regression, and this trend is weakened by adding a quadratic term in the regression, suggesting the latter is a better modeling choice; this type of exploration using only $\{Y_i, X_i\}$ is permitted without violating error control, and can be repeated as $\{A_i\}$ are revealed one by one. The power can be improved using the adjusted (quadratic) model because the i-bet test permits the analyst to explore models. For a fair comparison, the CovAdj Wilcoxon test is also implemented with a quadratic term.

### D.3. Wealth path in the real-data application

In the real-data application in Section 3, we report the final wealth after betting for all participants as $10^{15}$. Here we show the path of log-wealth $\log(M_t)$ for each iteration $t$ in Figure 6. The increase in wealth comes from correct guesses of the masked treatment assignments based on the outcomes and covariates (and the treatment assignments of the revealed subjects). We cumulate large wealth because most of the treatment assignments can be guessed correctly, indicating the treatment has an effect. The $p$-value is the inverse of the maximum wealth, which equals $10^{-16}$.

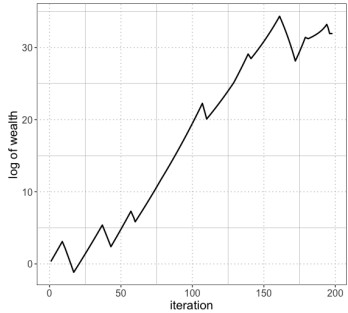

Figure 6: Path of the log of wealth against time, where an increase in the wealth indicates a correct guess of the treatment assignment. The wealth first fluctuates because our prediction of the masked treatment assignments is based on a few subjects with revealed assignments. Then, the wealth steadily increases because most assignments can be well-predicted. The wealth fluctuates at the end possibly because the remaining subjects do not have treatment effect, and hence we cannot guess their treatment assignments with high confidence.

## Appendix E. Extensions of i-bet to various other experimental settings

### E.1. Two-sample comparison with paired data

Suppose there are $n$ pairs of subjects. Let the outcomes of subjects in the $i$-th pair be $Y_{ij}$, the treatment assignments be indicators $A_{ij}$, the covariates be vector $X_{ij}$ for $j = 1, 2$ and $i \in [n]$. The null hypothesis of interest is that there is no difference between treatment and control outcomes conditional on covariates:

$$(Y_{ij} \mid A_{ij} = 1, X_{ij}) \overset{d}{=} (Y_{ij} \mid A_{ij} = 0, X_{ij}) \text{ for all } j = 1, 2 \text{ and } i \in [n]. \tag{23}$$

Consider a simple case of randomized experiments where

(i) the treatment assignments are independent across pairs, and randomized within each pair:

$$\mathbb{P}(A_{i1} = 1, A_{i2} = 0) = \mathbb{P}(A_{i1} = 0, A_{i2} = 1) = 1/2, \text{ for all } i \in [n];$$

(ii) the outcome of one subject $Y_{i_1, j_1}$ is independent of the treatment assignment of another subject $A_{i_2, j_2}$ for any $(i_1, j_1) \neq (i_2, j_2) \in [n] \times [2]$.

Under the null, observe that

$$\mathbb{P}(A_{i1} - A_{i2} = 1 \mid Y_{i1}, Y_{i2}, X_{i1}, X_{i2}) = 1/2 \quad \text{for all } i \in [n], \tag{24}$$

which implies the independence between $A_{i1} - A_{i2}$ and all outcomes and covariates. We can compress the paired data to an "unpaired" form, by treating the difference of paired assignments (after rescaling) $\widetilde{A}_i := (A_{i1} - A_{i2} + 1)/2$ as the pseudo treatment assignment, and the difference in the paired outcomes $\widetilde{Y}_i := Y_{i1} - Y_{i2}$ as the pseudo outcome, and the union of the covariates as the pseudo covariates $\widetilde{X}_i := \{X_{i1}, X_{i2}\}$. In such a way, algorithm 2 can be applied with pseudo data $\{\widetilde{Y}_i, \widetilde{A}_i, \widetilde{X}_i\}_{i=1}^n$, and guarantee valid error control for paired data. Meanwhile, under the alternative

with positive (negative) effect, the outcome difference $\widetilde{Y}_i$ is positively (negatively) correlated with the (rescaled) assignment difference $\widetilde{A}_i$, so our proposed tests can have nontrivial power. For example, in the i-bet test, the outcome difference $\widetilde{Y}_i$ can be used along with the union of covariates $\widetilde{X}_i$ to gather pairs with positive $\widetilde{A}_i$, as described in Algorithm 2 once we replace the input data with $\{\widetilde{Y}_i, \widetilde{A}_i, \widetilde{X}_i\}_{i=1}^n$.

Interestingly, we can derive another set of corresponding tests for the paired data from a different perspective. Rosenbaum (2002) and Howard and Pimentel (2020) consider the treatment-minus-control difference of the outcome, denoted as $D_i := (A_{i1} - A_{i2})(Y_{i1} - Y_{i2})$. Observe that under the null,

$$\mathbb{P}(\text{sign}(D_i) = 1 \mid |D_i|, X_{i1}, X_{i2}) = 1/2 \quad \text{for all } i \in [n], \tag{25}$$

because $(A_{i1} - A_{i2})$ has equal probability to be positive or negative as in (24). Note that here, we assume the outcomes are continuous to avoid nonzero probability of $\text{sign}(D_i) = 0$. Under the alternative, the treatment-minus-control difference $D_i$ can bias to positive (or negative) value. Therefore, all the discussed methods can be applied to the data $\{|D_i|, \text{sign}(D_i), \widetilde{X}_i\}_{i=1}^n$ where $\text{sign}(D_i)$ is viewed as the pseudo treatment assignment (if rescaled), and $|D_i|$ as the pseudo outcome.

### E.2. Multi-sample comparison *without* block structure

In multi-sample comparison, the case where subjects are not matched is often referred to as data without block structure, for which a classical test is the Kruskal-Wallis test (Kruskal and Wallis, 1952). We call the interactive test in this setting the i-Kruskal-Wallis test. Follow the notation of two-sample comparison with unpaired data in the previous section, where the treatment assignment $A_i$ now takes values in $[k] \equiv \{1, \ldots, k\}$ for $k$-sample comparison. The null hypothesis asserts that there is no difference between outcomes of any two treatments conditional on covariates:

$$(Y_i \mid A_i = a_1, X_i) \stackrel{d}{=} (Y_i \mid A_i = a_2, X_i) \text{ for all } i \in [n] \text{ and } a_1, a_2 \in [k]. \tag{26}$$

Consider a simple case with a randomized experiment, where we assume that

(i) the treatment assignments are independent and randomized

$$\mathbb{P}(A_i = a \mid X_i) = 1/k \text{ for all } i \in [n] \text{ and } a \in [k];$$

(ii) the outcome of one subject $Y_{i_1}$ is independent of the assignment of another $A_{i_2}$ for any $i_1 \neq i_2 \in [n]$.

Before introducing the interactive test, we first briefly describe the classical Kruskal-Wallis test.

**The Kruskal-Wallis test** The Kruskal-Wallis test considers the ranks of all observations. For subjects with treatment $a$, let the sample size be $N_a = \sum_{i=1}^n \mathbb{1}(A_i = a)$ and the average rank be $\overline{RK(a)} = \frac{1}{N_a} \sum_{i=1}^n \text{rank}(Y_i)\mathbb{1}(A_i = a)$. Denote the overall averaged rank as $\overline{RK} = \frac{1}{n} \sum_{i=1}^n \text{rank}(Y_i)$. The test statistic is

$$H = (n-1)\frac{\sum_{a=1}^k N_a \left(\overline{RK(a)} - \overline{RK}\right)^2}{\sum_{i=1}^n \left(\text{rank}(Y_i) - \overline{RK}\right)^2}, \tag{27}$$

which measures the relative variation across blocks and is expected to be large under the alternative. Thus, the Kruskal-Wallis test rejects the null if $H$ is larger than a threshold. The threshold is obtained from the null distribution of $H$, which can be derived if the sample size is small; otherwise, it is approximated by a chi-squared distribution.

**An interactive Kruskal-Wallis test**  The interactive test for multi-sample comparison is similar to the case of two-sample comparison. Both cases have the critical property that under the null, $A_i$ is independent of $\{Y_i, X_i\}$ with a known distribution. A difference from comparing two samples is that under the alternative, the association between the outcome $Y_i$ and the treatment $A_i$ can have various patterns depending on the underlying truth. Here, we consider an example of the i-Kruskal-Wallis test that targets a specific type of alternative.

Given three treatments ($k = 3$), suppose we wish to target the alternative of increasing outcomes:

$$(Y_i \mid A_i = 1, X_i) \preceq (Y_i \mid A_i = 2, X_i) \preceq (Y_i \mid A_i = 3, X_i), \tag{28}$$

where $Y^1 \preceq Y^2$ means that $Y^1$ is stochastically smaller than $Y^2$. The i-Kruskal-Wallis test can then use $\{Y_i, X_i\}$ to bet on whether $A_i$ is larger than its expected value $\mu_i = \mathbb{E}(A_i \mid X_i) = 2$. The complete procedure follows Algorithm 2 with $\mu_i = 2$ for all $i \in [n]$ and $w_t \in [-1, 1]$. Under the null, $M_t$ is a nonnegative martingale regardless of the bets and the error control is guaranteed.

### E.3. Multi-sample comparison *with* block structure

Suppose we want to compare $k$ treatments with $n$ blocks of data; a "block" is a group of $k$ subjects each of whom receives a different treatment (each treatment is assigned to exactly one subject). A classical test is the Friedman test (Friedman, 1937), and we call the interactive test as the i-Friedman test. For block $i \in [n]$ and subject $j \in [k]$, denote the outcome as $Y_{ij}$, the treatment assignment as $A_{ij}$, and the covariates as $X_{ij}$. The null hypothesis states that there is no difference between the outcome of any two treatments conditional on covariates:

$$(Y_{ij} \mid A_{ij} = a_1, X_{ij}) \overset{d}{=} (Y_{ij} \mid A_{ij} = a_2, X_{ij}) \text{ for all } j \in [k] \text{ and } i \in [n] \text{ and } a_1, a_2 \in [k]; \tag{29}$$

Consider a simple case of the randomized experiments where

(i) the treatment assignment $A_{ij}$ takes value $1, \ldots, k$ such that (a) $\{A_{i1}, \ldots, A_{ik}\}$ is equally likely to be any permutation of $\{1, \ldots, k\}$, and (b) the treatment assignments are independent across blocks;

(ii) the outcome of one subject $Y_{i_1, j_1}$ is independent of the assignment of another subject $A_{i_2, j_2}$ for any $(i_1, j_1) \neq (i_2, j_2) \in [n] \times [k]$.

**The Friedman test**  The Friedman test considers the ranks *within* each block $\{Y_{i1}, \ldots, Y_{ik}\}$, denoted as $\text{rank}(Y_{ij})$. Let the rank of the subjects with treatment $a$ averaged over $n$ blocks be $\overline{RK(a)} = \frac{1}{n} \sum_{i=1}^{n} \sum_{j=1}^{k} \text{rank}(Y_{ij}) \mathbb{1}(A_{ij} = a)$, and its expected value under the null is $\frac{1+k}{2}$. Under the alternative, the outcomes for one of the treatment could be larger (or smaller) than those for other treatments and the averaged rank would be higher (or lower). The Friedman test computes:

$$F = \sum_{a=1}^{k} \left( \overline{RK(a)} - \frac{1+k}{2} \right)^2,$$

and reject the null if $F$ is larger than a threshold obtained by the null distribution of $F$, which is approximated by a chi-square when $n$ or $k$ is large.

**An interactive Friedman test**  The interactive test for multi-sample comparison with block structure integrates the data within each block, similar to the case of paired sample for two-sample comparison. Consider the vector of treatment assignments within each block $i$ ordered by the outcomes, denoted as $\mathbf{A}_i = \big(A_{i,(1)}, \ldots, A_{i,(k)}\big)$, where $Y_{i,(1)} \geq \ldots \geq Y_{i,(k)}$. Because the assignments are independent of the outcomes under the null, we claim that

$$\mathbb{P}(\mathbf{A}_i = \mathbf{a} \mid \{Y_{ij}, X_{ij}\}_{j=1}^k) = 1/k! \quad \text{for all } \mathbf{a} \in \text{permute}([k]) \text{ and } i \in [n], \tag{30}$$

where permute($[k]$) denotes the set of all possible permutations of $[k]$. Under the alternative, the conditional distribution of $\mathbf{A}_i$ can bias to a certain ordering depending on the underlying truth.

As an example to compare three treatments ($k = 3$), suppose we wish to detect the following alternative:

$$(Y_{ij} \mid A_{ij} = 1, X_{ij}) \succeq (Y_{ij} \mid A_{ij} = 2, X_{ij}) \succeq (Y_{ij} \mid A_{ij} = 3, X_{ij}), \tag{31}$$

in which case $\mathbf{A}_i$ are more likely to be $(1, 2, 3)$. To develop an interactive test, we encode the vector of assignments by a scalar (pseudo assignment $\widetilde{A}_i$) such that it takes larger value when $\mathbf{A}_i$ is more "similar" to the ideal permutation $(1, 2, 3)$. Specifically, the similarity (distance) between $\mathbf{A}_i$ and $(1, 2, 3)$ can be measured by the number of exchange operations needed to convert $\mathbf{A}_i$ to $(1, 2, 3)$. We define $\widetilde{A}_i$ as:

$$\widetilde{A}_i = \begin{cases} 1, & \text{if } \mathbf{A}_i = (1, 2, 3), & (32) \\ 1, & \text{if } \mathbf{A}_i = (2, 1, 3), & (33) \\ 1, & \text{if } \mathbf{A}_i = (1, 3, 2), & (34) \\ -1, & \text{if } \mathbf{A}_i = (3, 1, 2), & (35) \\ -1, & \text{if } \mathbf{A}_i = (2, 3, 1), & (36) \\ -1, & \text{if } \mathbf{A}_i = (3, 2, 1), & (37) \end{cases}$$

where the ordered assignments (33) and (34) need one exchange operation to be converted to $(1, 2, 3)$; (35) and (36) need two; and (37) is the opposite of the ideal permutation, which needs three exchange operations. This design of $\widetilde{A}_i$ takes binary values, but it can also take different values for each ordering of $\mathbf{A}_i$. We present the above definition because it has a simple form and leads to relatively high power for a broad range of alternatives in simple simulations.

With the above transformation from a vector of assignments to a scalar $\widetilde{A}_i$ for each block $i$, we can view the blocks as individuals in the interactive test. That is, we use the pseudo assignment $\widetilde{A}_i$ for testing while ordering the blocks using the revealed data $\{Y_{ij}, X_{ij}\}_{i=1, j=1}^{i=n, j=k}$ and the actual assignments $\{A_{ij}\}_{j=1}^k$ once block $i$ is ordered. In other words, let the pseudo assignment $\widetilde{A}_i$ be defined in (32)-(37), the pseudo outcome be the union within each block, $\widetilde{Y}_i = \{Y_{ij}\}_{j=1}^k$, and same for the pseudo covariates $\widetilde{X}_i = \{X_{ij}\}_{j=1}^k$. The i-Friedman test follows Algorithm 2 with the input data replaced by $\{\widetilde{Y}_i, \widetilde{A}_i, \widetilde{X}_i\}_{i=1}^n$ and $\mu_i$ by 0 for all $i \in [n]$.

## E.4. Sample comparison in dynamic settings

We have proposed interactive tests for two/multi-sample comparison with unpaired/paired data, all of which are in the batch setting where the sample size is fixed before testing. Nonetheless, in many applications, one hopes to monitor the null of zero treatment effect as more subjects are collected,

so that the experiment can stop once there is enough evidence to reject the null. In this section, we consider a sequential setting where an unknown and potentially infinite number of subjects (or pairs) arrive sequentially in a stream and introduce the sequential interactive tests.

As a demonstration, we propose the seq-bet test for a two-sample comparison with unpaired data. Because the subjects arrive one by one, it is hard to order them on the fly, and we instead propose to filter the subjects to be cumulated in the product $M_t$. At time $t + 1$ when a new subject arrives, the analyst can interactively decide whether to include $A_{t+1}$ in current $M_t$. Denote the decision by an indicator $I_{t+1}$, and the product is

$$M_t = \prod_{i=1}^{t} \left[1 + I_i \cdot w_i \cdot (A_i - \mu_i)\right]. \tag{38}$$

The available information to decide $I_{t+1}$ and weight $w_{t+1}$ includes the complete data information of the first $t$ subjects and the revealed data of the $(t + 1)$-th subject, denoted by the filtration:

$$\mathcal{G}_t = \sigma\left(\{Y_i, A_i, X_i, I_i\}_{i=1}^{t} \cup \{Y_{t+1}, X_{t+1}\}\right), \tag{39}$$

where the complete data $\{Y_i, A_i, X_i\}_{i=1}^{t}$ can be used for modeling and guide the decision of $I_{t+1}$. Under the null, we have

$$\mathbb{P}(A_t = 1 \mid \mathcal{G}_{t-1}, I_t = 1) = \mu_t, \tag{40}$$

so the product $M_{t+1}$ is a martingale. Also, the martingale is nonnegative with bets in the range $w_i \in [-\frac{1}{1-\mu_i}, \frac{1}{\mu_i}]$. Thus, with the same argument as in Appendix A for the batch setting, Algorithm E.4 has valid error control as it stops and rejects the null when $M_t$ reaches the boundary $1/\alpha$.

**Input:** First sample $\{Y_1, A_1, X_1\}$, target type-I error rate $\alpha$;
**Procedure: for** $t = 1, 2, \ldots,$ **do**
    1. Using $\mathcal{G}_{t-1}$ to decide $I_t$, that is whether to include the $t$-th subject;
    2. Reveal $A_t$ and update $\mathcal{F}_t$;
    **if** $\prod_{i=1}^{t} \left[1 + I_i \cdot w_i \cdot (A_i - \mu_i)\right] > 1/\alpha$ **then**
      |  reject the null and stop;
    **else**
      |  Collect the $(t + 1)$-th sample $\{Y_{t+1}, A_{t+1}, X_{t+1}\}$
    **end**
**end**

  **Algorithm 4:** Framework of the sequential bet test (seq-bet)

In practice, to get a reasonably good model for our filtering process, we can first collect 50 subjects (say) and reveal their complete data $\{Y_i, A_i, X_i\}$ for modeling and then apply the seq-bet test from the 51-th subject. Note that Algorithm E.4 also applies to the sequential setting with paired data or multi-sample comparison when we replace the input data by pseudo sample $\{\widetilde{Y}_t, \widetilde{A}_t, \widetilde{X}_t\}$ defined in previous sections.

## Appendix F. Related work on weak null hypotheses

There are many works that focus on a less strict null hypothesis $H_0'$ than our global null $H_0$ in (1), which of course has pros and cons. These related methods would continue valid for the global null

hypothesis of our interest, but they could have lower power especially when $H_0'$ is true and $H_0$ is not true. Our strong global null is still sometimes of scientific interest, for example when certain quantiles of the distribution may be different under two different treatments (without the means differing), or one may be interested in the heavy-tailed case when the means may not even exist. We elaborate on the related work as follows.

While several works study treatment with multiple levels, for simplicity we describe them in the case with two levels (treated or not) in our discussion below. Akritas et al. (2000) assess the treatment effect by comparing the outcome CDF of treated and control group, denoted as $F_x^T(y)$ and $F_x^C(y)$ where $x$ is the given covariate value. Let $G(x)$ be a prespecified distribution for the covariate or its empirical distribution. The null hypothesis concerns marginal CDF after averaging over the covariate:

$$H_0' : \int F_x^T(y)dG(x) = \int F_x^C(y)dG(x), \tag{41}$$

which is implied by the global null $H_0$ in our discussion. Fan and Zhang (2017) also study the above null hypothesis (41), and propose an alternative test statistic to incorporate covariates. Wang and Akritas (2006) consider several extensions in the type of null hypothesis and suggest the possibility of testing whether the conditional outcome CDF given the covariates is identical:

$$H_0' : F_x^T(y) = F_x^C(y) \text{ for every } x \text{ and } y, \tag{42}$$

which is equivalent to the global null $H_0$ in our discussion, but no explicit test is provided for this null hypothesis. Similar null hypotheses are discussed in the work of Edgar Brunner (such as Akritas et al. (1997); Bathke and Brunner (2003)), which focus on factorial design and develop tests for the effect of one factor conditional on the level of the other factors. Thus, their methods can be used to test our global null $H_0$ when the covariate takes a finite number of values. Hettmansperger and McKean (2010) focus on testing the global null $H_0$ when the treatment effect is a linear function of the covariates, and discusses inference such as confidence intervals of the involved parameters. Along a different line of work, Thas et al. (2012) considers outcome $Y$ and covariates $Z$ (which include the treatment assignment $A$ and other covariates $X$ in our context) and let two instances $(Y, Z)$ and $(Y^*, Z^*)$ be independently distributed. The outcomes $Y$ and $Y^*$ are compared by estimating the probabilistic index $\mathbb{P}(Y > Y^* \mid Z, Z^*) + \frac{1}{2}\mathbb{P}(Y = Y^* \mid Z, Z^*)$. Their results imply a test for the null hypothesis of the probabilistic index being $1/2$, which can be used in our context:

$$\begin{aligned} H_0' : &\mathbb{P}(Y > Y^* \mid A = 1, A^* = 0, X = X^* = x) \\ &+ \tfrac{1}{2}\mathbb{P}(Y = Y^* \mid A = 1, A^* = 0, X = X^* = x) = 1/2 \text{ for all } x, \end{aligned}$$

which is true when our global null $H_0$ is true; hence, their method is valid for our problem of interest.

Aside from different target null hypotheses, several features distinguish our proposed algorithms from most existing work: (a) previous methods often commit to a single fixed procedure, while the i-bet test we propose can employ arbitrary working models, and the working model can be changed by a human analyst at any iteration to improve power; (b) most other methods mentioned above guarantee type-I error asymptotically, whereas our interactive methods have exact type-I error control (without any parametric or model assumptions on the outcomes); (c) we demonstrate through numerical experiments that the advantage of our proposed methods is more evident when a treatment effect exists only for a few subjects, whereas the above methods do not specifically focus on such sparse effects.

## Appendix G. Options for adjusting Wilcoxon's signed-rank test for covariates

The Wilcoxon signed-rank test is a simple and efficient nonparametric test with a known null distribution. Of course, rank-based statistics have been explored in many directions: see Lehmann and D'Abrera (1975) for a review of classical methods. Recent work focuses on how to incorporate covariate information to improve power. Zhang et al. (2012) develop an optimal statistic to detect constant treatment effect; in multi-sample comparison, Ding and Keele (2018) numerically compare rank statistics of outcomes or residuals from linear models; Rosenblum and Van Der Laan (2009) and Vermeulen et al. (2015) focus on related testing problems for conditional average effect and marginal effect; Rosenbaum (2010) and Howard and Pimentel (2020) use generalizations of rank tests for sensitivity analysis in observational studies. Here, we introduce variants of the signed-rank test for two-sample comparison in a randomized trial, which can improve the power of Rosenbaum's CovAdj Wilcoxon test under heterogeneous treatment effect.

The signed-rank test offers a general formula to construct tests for two-sample comparison. We note that the signed-rank test is perhaps more frequently used for paired data; but it can also be applied to unpaired data because the error control is also based on a decoupling between the sign and the rank. For each subject $i \in [n]$, let $E_i$ be any statistic that is larger when subject $i$ has treatment effect. We compute

$$W = \sum_{i=1}^{n} \text{sign}(E_i)\text{rank}(|E_i|), \tag{43}$$

and the null is rejected when $W$ is large. As an example, Rosenbaum (2002) proposed the covariance-adjusted signed-rank test by specifying $E_i$ as

$$E_i^{R(X)} := (2A_i - 1)R_i, \tag{44}$$

where recall $R_i$ is the residual of regressing $Y_i$ on $X_i$ without using $A_i$ as a predictor. (The covariance-adjusted signed-rank test is slightly different from the covariance-adjusted Wilcoxon rank-sum test (4), but they had similar power in most of our experiments.) The null distribution of $W$ depends on $E_i$, but one can use a permutation test that is valid for any choice of $E_i$, as described in Algorithm 1. Ideally, statistic $E_i$ should be designed to take a larger value when subject $i$ has a larger treatment effect. In the following, we discuss the question of whether the original choice of $E_i = E_i^{R(X)}$ can be improved, and which choice of $E_i$ should we prefer given different types of treatment effect.

### G.1. Existing statistics and their drawbacks

Aside from Rosenbaum's design of $E_i$ as $E_i^{R(X)}$, we can find several other alternatives to detect treatment effects in the causal inference literature. For example, one can construct a confidence interval for the ATE, which implies a test for zero ATE. However, the null of zero ATE is not the focus of this paper, as we are interested in the null of zero effect for any subpopulation. Lin (2013) suggests modeling $Y_i$ by a linear function of $A_i$ and $X_i$ (recently extended in a preprint by Guo and Basse (2021) to other parametric models), and construct the estimator for ATE as an average over subjects:

$$\frac{1}{n}\sum_{i=1}^{n}(2A_i - 1)(Y_i - \widehat{Y}(X_i; 1 - A_i)),$$

where $\widehat{Y}(\cdot;\cdot)$ denotes a fitted outcome using $X_i, A_i$ and $\widehat{Y}(X_i; 1 - A_i)$ predicts using the *false* assignment.

This estimator provides a design of $E_i$ that calculates the residual of predicting $Y_i$ using covariates $X_i$ and the false assignment $1 - A_i$ as follows:

$$E_i^{R(X,1-A)} := (2A_i - 1)(Y_i - \widehat{Y}(X_i; 1 - A_i)), \tag{45}$$

where $\widehat{Y}(X_i; 1 - A_i)$ can be the prediction via any black-box algorithm, such as a random forest.

There is also a rich literature on doubly-robust methods (see, for example, Robinson (1988); Robins et al. (1994); Cao et al. (2009); Chernozhukov et al. (2018)) to estimate ATE when the probability of receiving treatment varies with $X_i$. In a randomized experiment, the estimator boils down to

$$\frac{1}{n}\sum_{i=1}^{n}(2A_i - 1)(Y_i - \widehat{Y}(X_i; 1)/2 - \widehat{Y}(X_i; 0)/2),$$

which suggests a design of $E_i$ as $(2A_i - 1)(Y_i - \widehat{Y}(X_i; 1)/2 - \widehat{Y}(X_i; 0)/2)$. This design leads to similar power as $E_i^{R(X,1-A)}$ in most experiments and hence is omitted from this paper.

To examine the performance of tests using the statistics $E_i^{R(X)}$ and $E_i^{R(X,1-A)}$, we simulate outcomes from the generating model (2) where the function for treatment effect $\Delta$ and that for control outcome $f$ are constructed with different features (e.g., dense/sparse effect and bell-shaped/skewed control outcome):

$$\begin{aligned}
\Delta(X_i) &= S_\Delta\left[1 - |\sin(3X_i(3))|\right] && \textbf{(dense and weak effect)}; & (46)\\
\Delta(X_i) &= S_\Delta\left[2\exp\{X_i(3)\}\mathbb{1}\left(X_i(3) > 1.5\right)\right] && \textbf{(sparse and strong effect)}; & (47)\\
f(X_i) &= 5[X_i(1) + X_i(2) + X_i(3)] && \textbf{(bell-shaped control outcome)}; & (48)\\
f(X_i) &= 2\exp\{-2X_i(3)\}\mathbb{1}(X_i(3) < -2) && \textbf{(skewed control outcome)}. & (49)
\end{aligned}$$

The dense (sparse) effect is set to be weak (strong) since otherwise, all methods have power near one (zero).

We intentionally let the treatment effect and control outcome be nonlinear functions of the covariates because our discussion focuses on methods using nonparametric working models. In the rest of this paper, we employ random forests (with default parameters in the R package `randomForest`) as our working model since it usually generates good predictions for various data distributions (Breiman, 2001).

Although both methods have high power under a well-behaved distribution where the treatment effect is dense, the control outcome is bell-shaped, and the noise is standard Gaussian (solid lines in Figure 7(a)), they show different weak points when the effect is harder to detect—the test using $E_i^{R(X)}$ tends to have lower power when the treatment effect is sparse (Figure 7(b)); and the test using $E_i^{R(X,1-A)}$ tends to be less robust when the control outcome is skewed (Figure 7(c)). When the noise is heavy-tailed, both tests have lower power as expected, but the one using $E_i^{R(X,1-A)}$ appears to be more sensitive (Figure 7(a)). Broadly, the aforementioned pros and cons may be traced to two characteristics in the design of $E_i$:

(i) the prediction model that uses both $X_i$ and $A_i$ as in $E_i^{R(X,1-A)}$ accounts for heterogeneous treatment effect (by the interaction terms between $X_i$ and $A_i$), leading to high power for sparse effects;

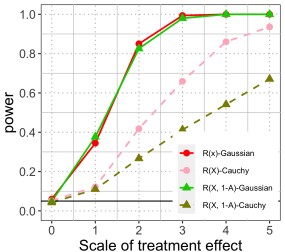
(*a*) Power when the treatment effect is dense and the control outcome is bell-shaped, and the noise varies as Gaussian and Cauchy (heavy-tailed).

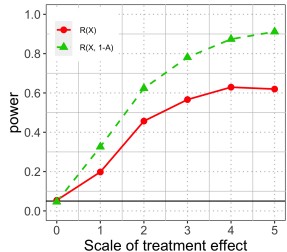
(*b*) Power when the treatment effect is sparse, the control outcome is bell-shaped, and the noise is Gaussian.

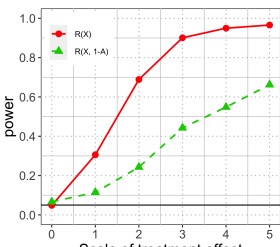
(*c*) Power when the treatment effect is dense, the control outcome is skewed, and the noise is Gaussian.

Figure 7: Power of the Wilcoxon test (43) using $E_i^{R(X)}$ and $E_i^{R(X,1-A)}$ as the scale of treatment effect $S_\Delta$ increases under different types of treatment effect, control outcome and noise. The test when using $E_i^{R(X,1-A)}$ tends to be more sensitive to heavy-tailed noise or skewed control outcome; and the test with $E_i^{R(X)}$ can have lower power when the treatment effect is sparse. Here and henceforth, we use 200 permutations, and the experiment is repeated 500 times.

(ii) the residuals in $E_i^{R(X)}$ only uses $X_i$ as predictors so that it effectively reduces the outcome variation that is *not* caused by the treatment, making the test robust under skewed control outcome.

Next, we propose other designs of $E_i$ that combine the advantages of the above two characteristics.

### G.2. Improve robustness under skewed control outcome by predicting residuals $R_i$

Because residuals $R_i$ can downsize the noise caused by skewed control outcome, we propose to measure the treatment effect via a prediction on $R_i$. That is, we compute the statistic $E_i$ by two steps of prediction:

(i) obtain residuals $R_i$ by predicting $Y_i$ using $X_i$ (without $A_i$);

(ii) fit a prediction model for $R_i$ using $X_i$ and $A_i$, denoted as $\widehat{R}(\cdot, \cdot)$;

(iii) get $E_i$ from the prediction error of $R_i$ using covariates $X_i$ and the false assignment $1 - A_i$:

$$E_i^{R-\widehat{R}(X,1-A)} := (2A_i - 1)(R_i - \widehat{R}(X_i, 1 - A_i)). \qquad (50)$$

Notice that $E_i^{R-\widehat{R}(X,1-A)}$ has a similar form as $E_i^{R(X,1-A)}$, where $\{R_i\}_{i=1}^n$ can be viewed as "denoised" outcomes: a large $Y_i$ could stem from skewness in the control outcome, but a large $R_i$ is

more likely to indicate large treatment effect, and hence achieves higher robustness to skewed control outcome. Numerical experiments coincide with our intuition: the power of using $E_i^{R-\widehat{R}(X,1-A)}$ improves from that using $E_i^{R(X,1-A)}$ when the control outcome is skewed (see Figure 8).

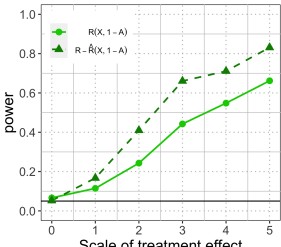 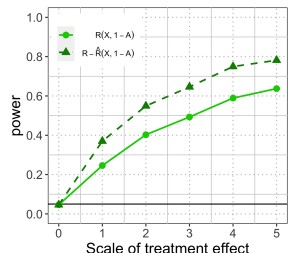

(*a*) Power when the treatment effect is dense and weak.

(*b*) Power when the treatment effect is sparse and strong.

Figure 8: Power of Wilcoxon test (43) using $E_i^{R(X,1-A)}$ and $E_i^{R-\widehat{R}(X,1-A)}$ as the treatment effect increases under skewed control outcome. The latter has higher power for both dense and sparse effects.

### G.3. Improve robustness under heavy-tailed noise using difference in the prediction error

Treating residuals $R_i$ as the pseudo outcomes is useful to account for variation in the control outcome, but $R_i$ can still contain much irrelevant variation, such as when the random noise $U_i$ in model (2) is Cauchy. Under heavy-tailed noise, the prediction model $\widehat{R}(\cdot, \cdot)$ in $E_i^{R-\widehat{R}(X,1-A)}$ could be inaccurate; and a large prediction error of using the false assignment as in $E_i^{R-\widehat{R}(X,1-A)}$ could result from heavy-tailed noise, while it is supposed to be evidence of large treatment effect.

So how to remove the large prediction error caused by poor modeling? We propose to consider the difference between the prediction error of using the false assignment $|\widehat{R}(X_i, 1 - A_i) - R(X_i)|$ and that using the true assignment $|\widehat{R}(X_i, A_i) - R(X_i)|$:

$$E_i^{|\widehat{R}(X,1-A)-R|-|\widehat{R}(X,A)-R|} := |\widehat{R}(X_i, 1 - A_i) - R(X_i)| - |\widehat{R}(X_i, A_i) - R(X_i)|. \tag{51}$$

Intuitively, when the prediction model $\widehat{R}(\cdot, \cdot)$ is a good fit, the prediction error using true assignment $|\widehat{R}(X_i, A_i) - R(X_i)|$ should be close to zero, and the proposed statistic is similar to $E_i^{R-\widehat{R}(X,1-A)}$. The advantage shows when the modeling is poor, such as under heavy-tailed noise. Here, the prediction error is large using either true or false assignment, so taking their difference as in $E_i^{|\widehat{R}(X,1-A)-R|-|\widehat{R}(X,A)-R|}$ can help rule out the variation caused by noise, letting the variation from treatment effect stand out. In the experiment with sparse effect (47), the test using $E_i^{|\widehat{R}(X,1-A)-R|-|\widehat{R}(X,A)-R|}$ has similar power as that using $E_i^{R-\widehat{R}(X,1-A)}$ when data is well-distributed (see Figure 9(*a*)), while it can achieve higher power under Cauchy noise or skewed control outcome (see Figure 9(*b*) and 9(*c*)), consistent with our intuition.

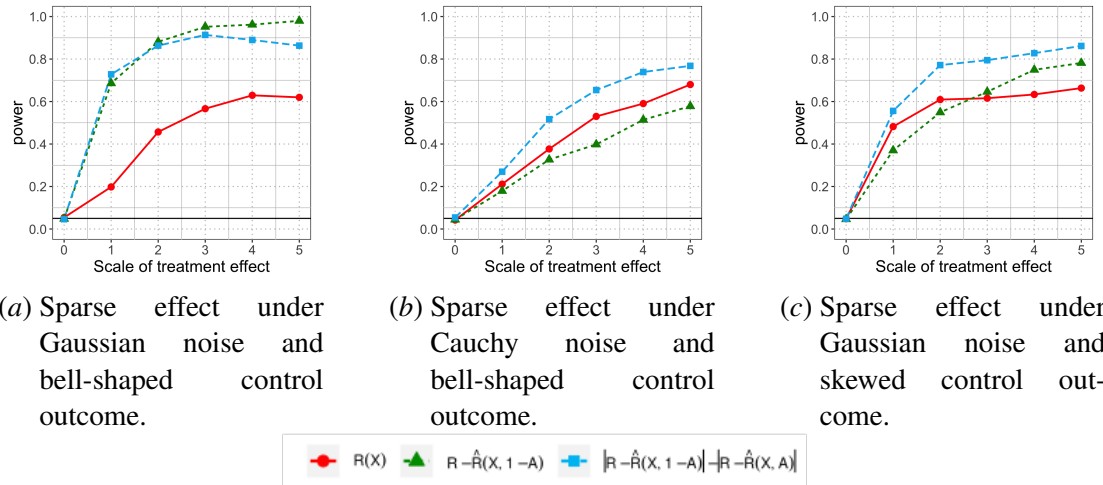

(*a*) Sparse effect under Gaussian noise and bell-shaped control outcome.

(*b*) Sparse effect under Cauchy noise and bell-shaped control outcome.

(*c*) Sparse effect under Gaussian noise and skewed control outcome.

Figure 9: The power of Wilcoxon test (43) using three statistics: $E_i^{R(X)}$, $E_i^{R-\widehat{R}(X,1-A)}$, and $E_i^{|\widehat{R}(X,1-A)-R|-|\widehat{R}(X,A)-R|}$ under sparse treatment effect, with the noise varies as Gaussian and Cauchy, and the control outcome varies as a bell-shaped or skewed distribution. The test using $E_i^{|\widehat{R}(X,1-A)-R|-|\widehat{R}(X,A)-R|}$ tends to have higher power especially under heavy-tailed noise or skewed control outcome.

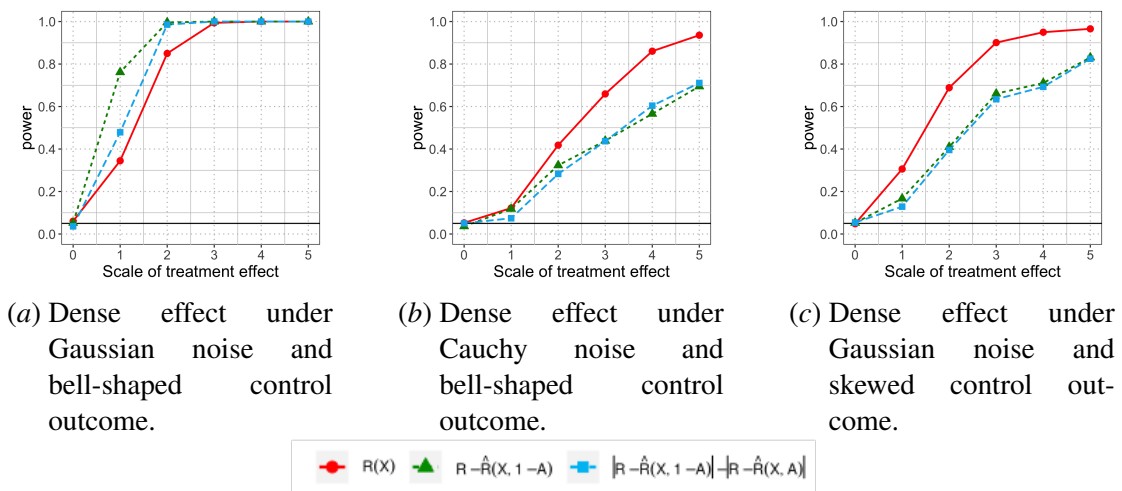

(*a*) Dense effect under Gaussian noise and bell-shaped control outcome.

(*b*) Dense effect under Cauchy noise and bell-shaped control outcome.

(*c*) Dense effect under Gaussian noise and skewed control outcome.

Figure 10: The power of Wilcoxon test (43) using three statistics: $E_i^{R(X)}$, $E_i^{R-\widehat{R}(X,1-A)}$, and $E_i^{|\widehat{R}(X,1-A)-R|-|\widehat{R}(X,A)-R|}$ under dense and weak treatment effect, when the noise varies as Gaussian and Cauchy, and the control outcome varies as a bell-shaped or skewed distribution. Rosenbaum's Wilcoxon test using $E_i^{R(X)}$ can be more robust to heavy-tailed noise or skewed control outcome.

**Remark 8** *Note that* $E_i^{|\widehat{R}(X,1-A)-R|-|\widehat{R}(X,A)-R|}$ *leads to high power when we want to detect a sparse and strong effect. However, when the effect is dense and weak as in model* (46)*, Rosenbaum's*

*Wilcoxon test using $E_i^{R(X)}$ is more robust to peculiar noise or control outcomes (see Figure 10). It is because $E_i^{|\widehat{R}(X,1-A)-R|-|\widehat{R}(X,A)-R|}$ uses a prediction model for $R_i$, which can be less informative for weak effect, especially when the noise is large. In practice, one may have some anticipation on the population properties of the treatment effect (density or strength), and choose the statistic accordingly. We summarize our recommendations under different settings in flowchart (56).*

### G.4. On one-sided versus two-sided effects

**The statistic of difference in the prediction error leads to high power for two-sided effects.** A major distinction between $E_i^{|\widehat{R}(X,1-A)-R|-|\widehat{R}(X,A)-R|}$ and the statistics discussed previously is that it takes large value for both positive and negative effects. It is because the difference in the prediction error of using opposite assignments is large as long as the assignment is a significant predictor for the outcome, regardless of the direction of effect. Therefore, the test using $E_i^{|\widehat{R}(X,1-A)-R|-|\widehat{R}(X,A)-R|}$ can cumulate effects of both signs while they cancel out in other statistics, leading to high power even when the average effect is close to zero. As some examples, we construct the following treatment effect:

$$\Delta(X_i) = S_\Delta \left[\exp\{X_i(3)\}\mathbb{1}\left(X_i(3) > 2\right) - X_i(1)/2\right] \tag{52}$$

**(Sparse strong positive effect and dense weak negative effect);**

$$\Delta(X_i) = S_\Delta \left[X_i^3(3)\mathbb{1}(|X_i(3)| > 1)\right] \tag{53}$$

**(Sparse strong effect of both signs);**

$$\Delta(X_i) = S_\Delta \left[\frac{2}{5}\sin(3X_i(3))\right] \tag{54}$$

**(Dense weak effect of both signs).**

In all examples, only the test using $E_i^{|\widehat{R}(X,1-A)-R|-|\widehat{R}(X,A)-R|}$ has nontrivial power (see the first row in Figure 11). Such sensitivity may or may not be desirable depending on the problem context. For example, we would hope to reject the null when the positive effect is strong for a subpopulation as in (52). However, one might want to treat a weak effect in both directions (54) as noise and leave the null unrejected. Next, we propose a modification of $E_i^{|\widehat{R}(X,1-A)-R|-|\widehat{R}(X,A)-R|}$ with such behavior.

**Targeting one-sided effects.** To differentiate between positive and negative effects, we modify the statistic $E_i^{|\widehat{R}(X,1-A)-R|-|\widehat{R}(X,A)-R|}$ by incorporating a sign that indicates the direction of the treatment effect. Consider the sign of two other statistics that approximate the treatment effect:

$$\begin{aligned}
S_i^1 &:= \mathbb{1}\{E_i^{R-\widehat{R}(X,1-A)} \geq 0\} \equiv \mathbb{1}\{(2A_i - 1)(R_i - \widehat{R}(X_i, 1 - A_i)) \geq 0\}, \\
S_i^2 &:= \mathbb{1}\{(2A_i - 1)(\widehat{R}(X_i, A_i) - \widehat{R}(X_i, 1 - A_i)) \geq 0\}, \quad \text{and combine them to get} \\
S_i &:= \mathbb{1}\{S_i^1 > 0 \text{ or } S_i^2 > 0\}.
\end{aligned}$$

We then define

$$E_i^{S\cdot(|\widehat{R}(X,1-A)-R|-|\widehat{R}(X,A)-R|)} := (2S_i - 1) \cdot E_i^{|\widehat{R}(X,1-A)-R|-|\widehat{R}(X,A)-R|}, \tag{55}$$

which is large when the treatment effect is large and *positive*. We tried using only $S_i^1$ or $S_i^2$ for the sign, but the combined one is more robust in experiments. The essential idea is to construct $S_i$ using some statistics that have a consistent sign with the treatment effect, while keeping the advantage of $E_i^{|\widehat{R}(X,1-A)-R|-|\widehat{R}(X,A)-R|}$ under skewed control outcome and heavy-tailed noise.

As desired, the test using $E_i^{S\cdot(|\widehat{R}(X,1-A)-R|-|\widehat{R}(X,A)-R|)}$ is less sensitive to weak effect of both signs (Figure 11(*c*)) and keeps high power for sparse strong positive effect (Figure 11(*d*)). Note that the signed statistic is more sensitive to noise because the signs are generated from less robust statistics (Figures 11(*e*), 11(*f*)). Nonetheless, among statistics that are insensitive to two-sided effect, $E_i^{S\cdot(|\widehat{R}(X,1-A)-R|-|\widehat{R}(X,A)-R|)}$ leads to high power for sparse effect, irrespective of whether the control outcome and the noise are well-distributed or have outliers.

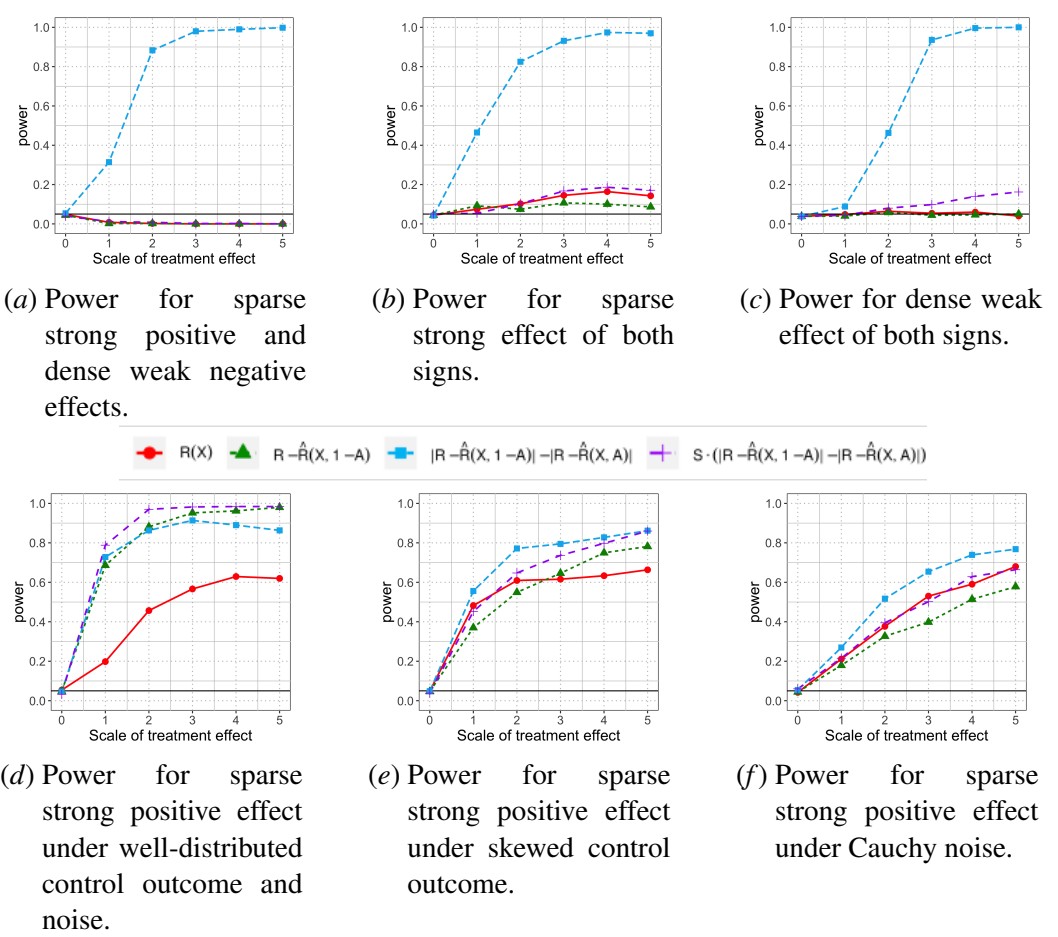

(*a*) Power for sparse strong positive and dense weak negative effects.

(*b*) Power for sparse strong effect of both signs.

(*c*) Power for dense weak effect of both signs.

(*d*) Power for sparse strong positive effect under well-distributed control outcome and noise.

(*e*) Power for sparse strong positive effect under skewed control outcome.

(*f*) Power for sparse strong positive effect under Cauchy noise.

Figure 11: Power of Wilcoxon test (43) using four statistics: $E_i^{R(X)}$, $E_i^{R-\widehat{R}(X,1-A)}$, $E_i^{|\widehat{R}(X,1-A)-R|-|\widehat{R}(X,A)-R|}$ and $E_i^{S\cdot(|\widehat{R}(X,1-A)-R|-|\widehat{R}(X,A)-R|)}$. In the first row where the treatment effect can be positive or negative, only the test using $E_i^{|\widehat{R}(X,1-A)-R|-|\widehat{R}(X,A)-R|}$ has nontrivial power. In the second row, the treatment effect is sparse and positive, and the control outcome and noise vary. The test using $E_i^{S\cdot(|\widehat{R}(X,1-A)-R|-|\widehat{R}(X,A)-R|)}$ can have high power without being too sensitive to the weak effect in both directions (see subplot 11(*c*)).

### G.5. Summarizing the observations made in this section

In this section, we proposed several variants of Rosenbaum's covariate-adjusted Wilcoxon as follows:

(i) Instead of predicting the *outcomes*, using the prediction model $\widehat{R}(\cdot, \cdot)$ for *residuals* $R_i$ can improve power under skewed control outcome. This is because the residuals $R_i$, which are themselves obtained by regressing $Y_i$ only on $X_i$ (without $A_i$), can remove much variation caused by the control outcome, and in turn highlight the treatment effect (see Appendix G.2).

(ii) The evidence of treatment effect can be measured by the prediction error using the false assignment, but a large prediction error could also be a result of a poorly fit model, such as when the noise is heavy-tailed. In contrast, the difference in the prediction error of using true and false assignments can eliminate most of the prediction error that is irrelevant to the treatment, including that from poorly fit models, and thus improve the power (see Appendix G.3).

(iii) The difference in prediction error detects both positive and negative effects with no distinction, so it can arguably be *too sensitive* (if there is such a thing) to a weak effect in both directions. If one wishes to target one-sided effects while maintaining the robustness achieved by "difference in prediction error", we propose to multiply it with an estimated sign of the effect (see Appendix G.4).

In summary, we recommend choosing one out of the three test statistics discussed in this section—$E_i^{R(X)}$, $E_i^{|\widehat{R}(X,1-A)-R|-|\widehat{R}(X,A)-R|}$, and $E_i^{S\cdot(|\widehat{R}(X,1-A)-R|-|\widehat{R}(X,A)-R|)}$—depending on one's prior belief of the population properties of treatment effect (if one exists), as shown below:

$$
\text{Nonzero effect} \begin{cases} \text{Effect of both signs} \rightarrow E_i^{|\widehat{R}(X,1-A)-R|-|\widehat{R}(X,A)-R|} \\ \text{Positive effect} \begin{cases} \text{Sparse and strong effect} \rightarrow E_i^{S\cdot(|\widehat{R}(X,1-A)-R|-|\widehat{R}(X,A)-R|)} \\ \text{Dense and weak effect} \rightarrow E_i^{R(X)} \end{cases} \end{cases}
$$

$$(56)$$

Note that the i-bet test is not included here because its performance depends on the interaction and progressive updates to the initial working model made by the analyst based on revealed data. The flexibility makes the i-bet test a potentially more robust and promising method compared with the aforementioned methods that also use a parametric (or semiparametric) working model.

