# OpenReview forum: "Interactive rank testing by betting"
_cclear.cc/CLeaR/2022/Conference — CLeaR 2022 Oral_

### Official Review · Reviewer_GSiN · 2021-11-22

**Confidence:** 2
**Overall Score:** 7

**Main Review:**

Originality: This paper provides a novel interactive betting hypothesis testing framework. Its novelty is justified in my opinion,and I'm not aware of existing work that already solves this problem.

Significance: The problem of detecting the causal effect of a treatment is one that is important in many fields in the natural sciences, social sciences and engineering. This proposed method gives a game theoretic insight into the problem of hypothesis testing.

Technical quality: To my best understanding, the claims are theoretically sound and substantiated by theorems put forth in the paper.

Clarity: The submission is clearly written as well as a comparison to existing methods is made and the performance demonstrated on simulated data.

**Summary:**

This paper proposes a novel interactive hypothesis testing framework that possesses desirable statistical properties as well as computational advantages over existing permutation based tests. The theoretical properties are proved and performance is demonstrated on simulated data.

---

> ### Author Response · Authors · 2021-12-04
> **Response to Reviewer GSiN**
>
> We thank the reviewer for the positive comments.

---

### Official Review · Reviewer_zjzR · 2021-11-24

**Confidence:** 2
**Overall Score:** 6

**Main Review:**

The key idea in this paper is summarized in Remark 2,  that whenever the test statistic is a nonnegative martingale (by design), permutations can be avoided. Based on this observation and Ville's (Doob's) maximal inequality, the authors construct a process M_t and show that it is a nonnegative martingale under the null.

I am not very familiar with the interactive testing literature, but at least the application of this idea for testing the global null hypothesis appears to be new.

1. In equation (1), it is unclear what the notation Y_i | A_i=1, X_i refers to. According to the context, this should refer to the potential outcomes, but the notation seems to suggest observed outcomes.

2. The authors describe a specific way of choosing \pi_t in Section 2.2. I am wondering if they could comment on how would different choices of \pi_t would influence the power of the proposed test, and how should a practitioner make such choices in a real setting.

3. It is a pity that they did not apply this test to a real data set. Such data sets seem to be readily available, and a data application would help illustrate the method better, and help clarify such practice choices needed to made by practitioners; e.g. see my comment 2 above.

**Summary:**

The paper proposes a non-parametric tests for testing the global null hypothesis in a randomized experiment. It avoids the permutations employed in traditional nonparametric tests by interactive testing, that was recently introduced in the literature.

---

> ### Author Response · Authors · 2021-12-04
> **Response to Reviewer zjzR**
>
> We thank the reviewer for the comments. Below are our responses.
>
> > In equation (1), it is unclear what the notation $Y_i | A_i=1$, $X_i$ refers to. According to the context, this should refer to the potential outcomes, but the notation seems to suggest observed outcomes.
>
> To answer the review's question, we provide an alternative, equivalent, description in the language of observed and potential outcomes. Let's say that each subject $i$ has potential control outcome $Y_i^C$, potential treated outcome $Y_i^T$, and the treatment indicator~$A_{i}$ for $i \in [n] \equiv \{1,2,\dots,n\}$. The observed outcome is $Y_i = Y_i^C(1 - A_i) + Y_i^T A_i$ under the standard causal assumption of consistency ($Y_i = Y_i^T$ when $A_i = 1$ and $Y_i = Y_i^C$ when $A_i = 0$). In this setting, we have $Y_i | A_i=1$ is precisely $Y_i^T$. We have added the above description in Section 1.1.
>
> > The authors describe a specific way of choosing $\pi_t$ in Section 2.2. I am wondering if they could comment on how would different choices of $\pi_t$ would influence the power of the proposed test, and how should a practitioner make such choices in a real setting.
>
> The choice of the ordering $\pi_t$ affects the test power, when taken together with the choice of weight $w_t$. So let us first note that a desirable weight $w_t$ should ideally have the same sign as $A_{\pi_t}$; this would allow $M_t$ to increase to sooner reach the rejection threshold $1/\alpha$. Therefore, we recommend practitioners to order upfront the subjects for which they (or the algorithm acting on their behalf) are most confident about their treatment assignment.
>
> We have added the above recommendation as a remark in Section 2.1.
>
> > It is a pity that they did not apply this test to a real data set. Such data sets seem to be readily available, and a data application would help illustrate the method better, and help clarify such practical choices needed to made by practitioners; e.g. see my comment 2 above.
>
> We have now run the methods on 4 real-world datasets, in 3 all of them don't reject (data in [1], [2], and the nudge-intervention experiment in [3]), and in the fourth all of them reject (data in the voucher-intervention experiment in [3]). Hence we continue to focus more on the simulations where the ground truth is known, in order to compare the methods in detail. The reason we did not add the real-data application earlier is that we cannot compare the power by doing repeated simulations; we can run each test once and get a binary signal about whether they are rejected or not. Also, since we do not know the ground truth, we would not even be sure if it is a false positive or not. Nevertheless, we will add some real data experiments to the paper, and mention the above caveats.
>
> As an example, the data in the last experiment is from Hainmueller et al. (2018) [3], which investigates whether application fees act as a barrier for low-income immigrants who want to become US citizens. A relevant part of their study collected $750$ participants who are supposed to pay the application fee at the time, and assigned them into two groups using complete randomization. The treated group ($n_1 = 210$) received a voucher that pays the fee for the naturalization application; while the control group ($n_0 = 540$) did not receive this voucher. The outcome $Y_i$ is whether or not participant $i$ reported having submitted the US citizenship application. The covariates $X_i$ include age, gender, years holding a green card, marital status, and educational attainment.
>
>
> ---
> [1] Daren R Anderson, Samantha  Horn, Dean Karlan, Amanda E Kowalski, Jody L Sindelar, and Jonathan Zinman. Evaluation of combined financial incentives and deposit contract intervention for smoking cessation: A randomized controlled trial. Journal of Smoking Cessation, 2021.
>
> [2] Manish Gehani, Suman Kapur, Sudha D Madhuri, Vara Prasad Pittala, Sree Kala Korvi, NagamaniKammili, and Shashwat Sharad. Effectiveness of antenatal screening of asymptomatic bacteriuria in reduction of prematurity and low birth weight:  Evaluating a point-of-care rapid test in a pragmatic randomized controlled study. EClinicalMedicine, 33:100762, 2021.
>
> [3] Jens Hainmueller, Duncan Lawrence, Justin Gest, Michael Hotard, Rey Koslowski, and David D Laitin. A randomized controlled design reveals barriers to citizenship for low-income immigrants. Proceedings of the National Academy of Sciences, 115(5):939–944, 2018.

---

### Official Review · Reviewer_URGT · 2021-11-24

**Confidence:** 3
**Overall Score:** 7

**Main Review:**

Strength:

- It follows a clear research direction from Wilcoxon rank test to CovAdj Wilcoxon test and to the proposed i-bet test.

- The idea to introduce reward and use a game-theoretic way to design the test is interesting. The intuition is explained clearly by "If an analyst can guess most treatment assignments correctly, then the treatment must have an effect".

- The cumulative products $M_t$ is well designed with good martingale properties under the null.

Room for improvements:

- Current simulation setting is relatively simple. The covariates have a dimension of 3 and follow simple distribution. It is better to evaluate in more challenging case.

- In the simulation, it is better to include more details such as how to compute the power in Figure 2 and 3 for the proposed method.

- Why in Fig. 2 linear-CATE-test is better than i-bet but in Fig. 3 it is much worse? How to choose which one to use?

- One writing suggestion is to have a minimal introduction to the full algorithm (e.g., alg. 2) and with a demonstrative example first. Current methodology is not introduced until page 6, with many discussions and related works discussed before.

In general, this paper presents an interesting idea and I recommend for acceptance.

Post-rebuttal: thanks for the authors' feedback and I would maintain my recommendation.

**Summary:**

This paper design an interactive test called i-bet. The paper deals with classic randomized experiments  The idea is to test by betting from a game-theory perspective, which improves the power under heterogeneous treatment effects.

---

> ### Author Response · Authors · 2021-12-04
> **Response to Reviewer URGT**
>
> We thank the reviewer for the comments. Below are our responses.
>
> > Current simulation setting is a relatively too simple. The covariates have a dimension of 3 and follow simple distribution. It is better to evaluate in more challenging case.
>
> The simulation was intended for illustration only, since we felt it was easier to highlight certain behaviors when it is clear what the test is doing. In a high-dimensional setting, with odd noise distributions, and a nonlinear signal, it is harder for us to reason about which test is failing/succeeding for what reason. In other words, the restriction is not in the applicability of the method, but in the human ability to comprehend and explain what the methods are doing. In Appendix D, we included more simulation settings on non-Gaussian heavy-tailed Cauchy noise and nonlinear signals.
>
> > In the simulation, it is better to include more details such as how to compute the power in Figure 2 and 3 for the proposed method.
>
> We have added the details of power computation under Figure 2. The power is averaged over 500 repetitions, and estimated by the proportion of the repeated experiments where the null gets rejected.
>
> > Why in Fig. 2 linear-CATE-test is better than i-bet but in Fig. 3 it is much worse? How to choose which one to use?
>
> In Figure 2, the linear-CATE-test has higher power because the simulated data for Figure 2 is generated such that the outcome is a linear function of the covariates (with interaction term) by equations (12) and (13), which is consistent with the assumed model in the linear-CATE-test. However, the linear-CATE-test cannot behave as well in Figure 3 because the simulated data using equation (14) does not follow a linear relationship. We recommend using the linear-CATE-test if the practitioners strongly believe the outcome indeed is a linear function of the covariates; otherwise, we would recommend our proposed i-bet test so that we can explore and choose the appropriate model as the test proceeds, for example starting with a linear model, and then re-evaluating that choice  (based on a few unmasked treatment assignments) if the first few bets are as good as random chance, instead of resulting in growing capital.
>
> To elaborate on the latter point briefly, we view the simulations with i-bet as a thought experiment: the reader must imagine that we perhaps chose a poor model for all methods at the start. Using a poor (uninformative or barely better than chance) model, our bet $w_j$ would possibly have a random sign for early subjects, and our wealth may fluctuate up and down, rather than increase reasonably steadily. The multi-step i-bet test can have higher power than other single-shot tests because, in the midst of this testing, the analyst can observe the poor start, explore and evaluate various models using the complete data for completed bets and the masked-assignment data for every other point, and try to find a better model. We mentioned the above note in "Illustrations of adaptive modeling" of Section 3; and to highlight the above idea, we have moved this discussion to the beginning of Section 3.
>
> > One writing suggestion is to have a minimal introduction to the full algorithm (e.g., alg. 2) and with a demonstrative example first. Current methodology is not introduced until page 6, proceed with many discussions and related works.
>
> In the original paper, we attempted to introduce the full algorithm as the reviewer suggested: see equation (5) for the test statistic, the follow-up description that the algorithm should stop when the statistic exceeds $1/\alpha$, and equation (6) for some intuition on the error control. Nevertheless, we can add a forward reference to the full algorithm in the above-mentioned paragraph, and also add an illustrative example.

---

### Decision · Program_Chairs · 2022-01-12

**Decision:**

Accept (Oral)

**Comment:**

The paper proposes a new test for determining whether a treatment has an effect, rooted in a game-theoretic framework.

The reviewers and area chair all liked the paper and the authors' rebuttal convincingly answers their questions.

Please follow the suggestions when writing the revised version of the paper (computing the test power; choosing $\pi_t$ and $w_t$) and discussing in more length how the approach can be assessed on real-world data, when of course the ground truth is unknown.